# Histone H3 binding to the PHD1 domain of histone demethylase KDM5A enables active site remodeling

James E. Longbotham[1], Cynthia M. Chio[1], Venkatasubramanian Dharmarajan[2], Michael J. Trnka[3], Idelisse Ortiz Torres[4], Devrishi Goswami[2], Karen Ruiz[5], Alma L. Burlingame[3], Patrick R. Griffin [2] & Danica Galonić Fujimori[1,3]

Histone demethylase KDM5A removes methyl marks from lysine 4 of histone H3 and is often overexpressed in cancer. The in vitro demethylase activity of KDM5A is allosterically enhanced by binding of its product, unmodified H3 peptides, to its PHD1 reader domain. However, the molecular basis of this allosteric enhancement is unclear. Here we show that saturation of the PHD1 domain by the H3 N-terminal tail peptides stabilizes binding of the substrate to the catalytic domain and improves the catalytic efficiency of demethylation. When present in saturating concentrations, differently modified H3 N-terminal tail peptides have a similar effect on demethylation. However, they vary greatly in their affinity towards the PHD1 domain, suggesting that H3 modifications can tune KDM5A activity. Furthermore, hydrogen/deuterium exchange coupled with mass spectrometry (HDX-MS) experiments reveal conformational changes in the allosterically enhanced state. Our findings may enable future development of anti-cancer therapies targeting regions involved in allosteric regulation.

[1] Department of Cellular and Molecular Pharmacology, University of California, 600 16th Street, Genentech Hall, San Francisco, CA 94158, USA. [2] Department of Molecular Medicine, The Scripps Research Institute, Jupiter, FL 33458, USA. [3] Department of Pharmaceutical Chemistry, University of California, 600 16th Street, Genentech Hall, San Francisco, CA 94158, USA. [4] Chemistry and Chemical Biology Graduate Program, University of California, 600 16th Street, Genentech Hall, San Francisco, CA 94158, USA. [5] TETRAD Graduate Program, University of California, 600 16th Street, Genentech Hall, San Francisco, CA 94158, USA. Correspondence and requests for materials should be addressed to D.G.F. (email: Danica.Fujimori@ucsf.edu)

Post-translational modifications of histone proteins are important regulators of chromatin structure and function and are controlled by proteins that write, read and erase these marks[1,2]. A common and functionally diverse histone modification is lysine methylation, which regulates many cellular processes, including heterochromatin formation, regulation of transcription and DNA repair[3,4]. Lysine methylation is a reversible modification, and its removal is catalyzed by lysine histone demethylases (KDMs). The KDMs are grouped into several subfamilies depending on their domain composition, substrate specificity and reaction mechanism. The KDM1 family (LSD1 and LSD2) uses a flavin-dependent mechanism, and acts on mono- or di-methylated lysines[5,6]. A broader range of demethylation is possible by the jumonji C (JmjC) domain-containing family of KDMs (KDM2-9) that utilize a Fe(II)- and α-ketoglutarate (α-KG)-dependent mechanism as they are able to demethylate mono-, di- and tri- methylated lysines[7]. They predominantly act on histone proteins, but in some instances also catalyze demethylation of non-histone substrates[8,9]. Understanding the role of the chromatin environment in regulating activities of these enzymes is critical to elucidation of context-dependent spatial and temporal regulation of chromatin methylation. Several reports in recent years have pointed out the critical role of chromatin reader domains in regulation of demethylase activities, substrate specificities, and localization[10–13].

The human KDM5 subfamily of JmjC demethylases consists of four family members, KDM5A-D, which demethylate H3K4me1/2/3 marks. The proteins in the KDM5 family share common structural features, such as an iron containing active site comprised of the JmjN and JmjC domains[14–16], a DNA binding ARID domain, a zinc-finger domain, and either two (for KDM5C and D) or three (for KDM5A and B) plant homeodomain (PHD) chromatin reader domains[7,17–19] (Fig. 1a). There has been a considerable amount of interest in the KDM5 family due to their roles in many disorders as all four members have been shown to be involved in various cancers[20–24]. Specifically, KDM5A is overexpressed in breast cancer[25] and its fragment is known to form a fusion with NUP98 in acute leukemia[20]. Additionally, there is evidence for overexpression of KDM5A in cancer drug resistance in lung cancer models[26] as well as osteoporosis[27]. KDM5B is overexpressed in hepatocellular carcinoma where it promotes metastasis[28]. Additionally, this enzyme is involved in drug resistance in melanoma treatments[29] and regulation of genes involved in stem cell differentiation[22,30]. KDM5C is highly expressed in neuronal tissues and mutations in this enzyme have been associated with X-linked intellectual disability disorders[17,31]. KDM5D has been suggested to have a role in spermatogenesis[32]. It is for these reasons that the KMD5 family are of clinical interest, prompting investigations into development of small molecule inhibitors of these enzymes[15,16,33–37].

KDM5A contains three PHD domains (Fig. 1a), which are commonly recognized as chromatin readers and are often found in chromatin modifying enzymes. PHD domains in demethylases have traditionally been associated with the recruitment of demethylases to chromatin[38]. For example, PHD domain protein BHC80 is a component of LSD1 co-repressor complex that stabilizes the recruitment of LSD1 to chromatin[39]. In addition, PHD domains can also regulate substrate specificity of demethylases, as in the case of PHF8 and KIAA1718[10]. In vitro studies on PHD domains in KDM5A have shown that PHD1 preferentially binds unmodified H3 N-terminal peptides[40] and that PHD3 binds H3K4me3 marks[20], while no function for PHD2 has yet been determined. Binding of the H3K4me3 mark by PHD3 enables the recruitment of the demethylase to its substrate, a role consistent with the prototypical function of reader domains. In contrast to this canonical role, our earlier in vitro studies on the PHD1 domain have uncovered a function of the PHD1 domain in the allosteric regulation of the demethylase activity of KDM5A[40]. Specifically, we found that the engagement of the PHD1 domain by unmodified H3 peptide enhances the activity of the catalytic domain. Since H3 peptide unmodified at Lys4 is the product of demethylation, these findings suggested a positive feedback

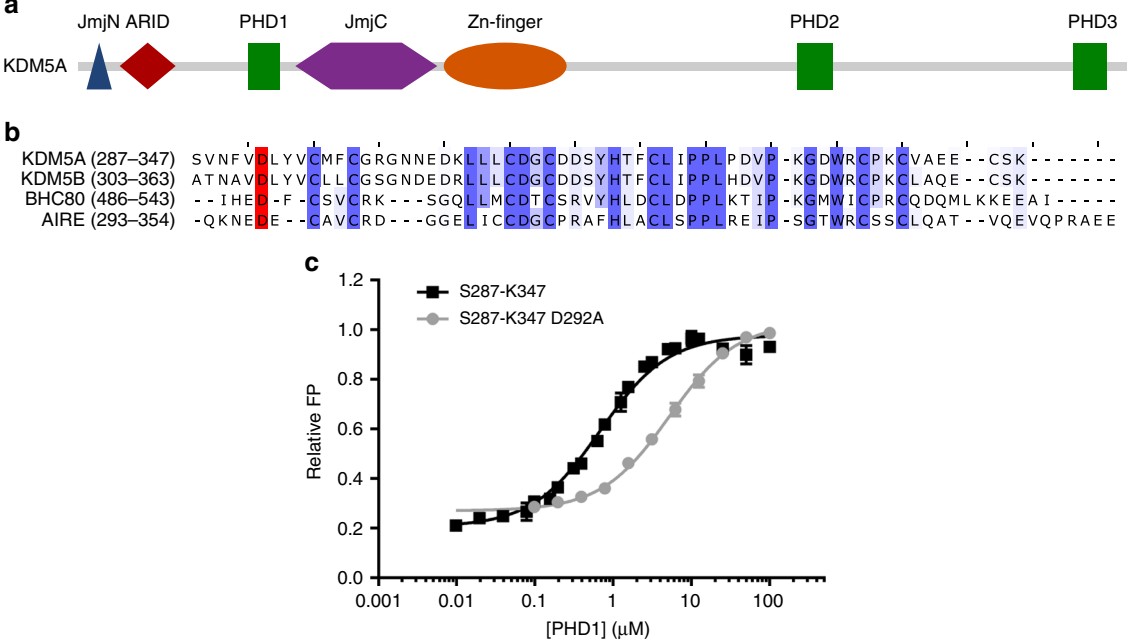

**Fig. 1** Asp292 is important for PHD1 binding to H3 N-terminal peptides. **a** Domain structure of KDM5A. **b** Sequence alignment of several PHD domains that preferentially bind unmethylated H3K4 peptides. The conserved Asp residues are highlighted in red. Other residues conserved between the different PHD domains are highlighted in blue. **c** Normalized fluorescence polarization of 5-carboxyfluorescein (5-FAM) conjugated H3 10mer peptide binding to PHD1 and PHD1 D292A. Data were fitted to equation 1 or 2 and binding parameters are shown in Table 1. Errors ($n \geq 3$) represent s.e.m.

regulatory mechanism that may allow for demethylation to be propagated along nucleosomes. Such feedback regulation could be important for removal of broad H3K4me3 domains in early embryo development, a process mediated by KDM5A[41] and KDM5B[42,43].

The molecular basis of allosteric stimulation by the product peptide binding to the PHD1 domain in KDM5A remains poorly understood. In addition, it remains unclear how the allostery is propagated between the PHD1 and JmjC domains. Current crystal structures of KDM5 enzymes are missing the structure of PHD1 and its surrounding regions due to either the PHD1 domain being removed from the construct or being disordered in these structures[14–16,35]. With limited structural information, here we aim to determine the mechanistic rationale for the catalytic enhancement that results from binding of the peptide ligand to the PHD1 domain. Using purified KDM5A, we demonstrate that this allosteric enhancement is caused by improved substrate binding rather than enhanced catalytic turnover. Differently modified H3 tail peptides bound to the PHD1 domain promote similar effects, indicating that affinity of the modified H3 ligand, which would affect occupancy of the PHD1 domain, drives ligand-induced allosteric stimulation. Hydrogen/Deuterium exchange MS (HDX-MS) was used to probe conformational changes and highlighted regions of the catalytic domain that become less solvent protected in the allosterically enhanced state.

## Results

**Asp292 is important for PHD1 in recognizing H3 peptides.** The allosteric regulation of KDM5A by its PHD1 domain highlights a more complex role of this domain rather than simple recognition of chromatin marks. Therefore, it is important to understand the molecular basis of how the PHD1 domain interacts with its ligand. PHD1$_{KDM5A}$ binds the N-terminal tail of histone H3 with a preference for unmodified H3, and binding affinity decreases with an increase in H3K4 methylation[40]. Structural investigations into PHD1$_{KDM5B}$ and several other PHD1 domains, which like PHD1$_{KDM5A}$ preferentially recognize unmodified H3K4 residues, demonstrate electrostatic interactions between the unmodified H3K4 residue and a conserved carboxylate residue[39,44–46] (Fig. 1b). However, our initial NMR studies of PHD1$_{KDM5A}$ have not detected any interactions between the equivalent conserved carboxylate residue (Asp292) and H3K4 in the histone peptide[40], necessitating further investigations into binding of H3K4 by PHD1 of KDM5A.

We suspected that the discrepancy regarding the role of the carboxylate residue in PHD1$_{KDM5A}$ and PHD1$_{KDM5B}$ could be due to the Asp292 residue in PHD1$_{KDM5A}$ being close to the N-terminus of the construct used in our previous studies (V291-K347 used in binding assays, D292-E344 used in NMR experiments). A new construct was generated (S287-K347) (Supplementary Figure 1) and used to measure binding affinity for the H3 peptide via a fluorescence polarization assay (Fig. 1c). A D292A variant was also generated to determine the contribution of the conserved Asp residue to peptide binding. Importantly, this mutation does not impair the overall fold or stability of the PHD1 domain (Supplementary Figure 2).

Compared to the PHD1$_{V291-K347}$ construct, the PHD1$_{S287-K347}$ construct shows an approximate twofold increase in affinity for the unmodified and mono-methylated, while di- and tri-methylated H3K4 peptides have similar binding affinity to both PHD1 constructs (Table 1). Our observations suggest that the additional N-terminal residues (S287-F290) are contributing to a higher affinity for unmodified and mono-methylated H3K4 peptides. This is further supported by the observation that only the longer PHD1$_{S287-K347}$ shows a 2.3-fold preference for WT H3

**Table 1 Binding affinities of differently modified H3 histone peptides for different PHD1 constructs**

|  | $K_d$ (μM) | | |
|---|---|---|---|
|  | V291-K347 PHD1 | S287-K347 PHD1 | S287-K347 PHD1 D292A |
| H3K4me0[a] | 1.89 ± 0.16 | 0.68 ± 0.04 | 5.20 ± 0.26 |
| H3K4me1 | 16.0 ± 3.0 | 9.49 ± 0.92 | 153 ± 33 |
| H3K4me2 | 15.8 ± 2.2 | 13.0 ± 1.7 | 141 ± 28 |
| H3K4me3 | 35.5 ± 7.5 | 34.0 ± 5.5 | 511 ± 281 |
| H3K4A | 2.11 ± 0.27 | 1.58 ± 0.19 | 22.2 ± 5.7 |

[a]Indicates affinities determined by direct FP

peptide over the H3K4A peptide, whereas the binding affinities of WT H3 and H3K4A peptides to the shorter PHD1$_{V291-K347}$ are highly similar (Table 1). Our findings with the extended construct are similar to the 2.5-fold discrimination of WT H3 over H3K4A peptide that has been previously observed for PHD1$_{KDM5B}$[44].

We further probed the PHD1-H3 interaction using a PHD1$_{S287-K347}$ D292A variant. Compared to the WT PHD1$_{S287-K347}$, the mutant PHD1 domain shows an approximately eightfold decrease in affinity for the unmodified H3 peptide (Fig. 1c, Table 1) and a 10–16-fold decrease in affinity for methylated H3K4 peptides (Table 1) highlighting importance of Asp292 in contributing to PHD1-H3 tail interaction. These large fold reductions in affinity suggest that contribution of D292 extends beyond its likely electrostatic interaction with Lys4 in H3 and into structural organization of the N-terminus of the PHD1 for histone binding. This conclusion is consistent with structural analysis of the interaction of the highly homologous PHD1$_{KDM5B}$ with H3, where the equivalent carboxylate residue facilitates recognition of H3T6[44]. Together, our findings suggest that the proper orientation of the Asp292 residue in the longer PHD1 construct significantly contributes to binding of the H3 peptide to the PHD1 domain.

**PHD1 occupancy improves the catalytic efficiency of KDM5A.** We have previously shown that binding of a H3 peptide to the PHD1 domain enhances the rate of demethylation by KDM5A in vitro[40]. To dissect contributions of changes in substrate binding and enzymatic turnover to the overall allosteric enhancement, we investigated how the Michaelis–Menten kinetic parameters were affected by ligand binding to the PHD1 domain. A challenge for such an investigation lies in the nature of the ligand for each of the two sites, the PHD1 domain and the catalytic domain. While the PHD1 domain preferentially interacts with unmodified H3, it also binds H3K4me3, the substrate of the catalytic domain, albeit with a decreased affinity (Table 1). This cross-binding between two sites first prompted us to develop a discrete set of peptide ligands for the two sites in KDM5A (Fig. 2a). After studying the available structural information, N-terminal acetylation of the H3 peptide would likely prevent its binding to the PHD1 domain, as is the case in PHD$_{UHRF1}$[46]. Indeed this was the case, as an N-terminally acetylated 21mer H3K4me2 peptide (Ala1-Ala21) does not bind to the PHD1 domain at the concentrations used in our assays (Supplementary Figure 3a). Importantly, although activity is reduced relative to corresponding peptide with a free N-terminus (Supplementary Figure 3c), N-terminal acetylation is tolerated by the catalytic domain (Fig. 2b, c). To develop a peptide selective for the PHD1 domain, we relied on the observation that C-terminal truncation of the K4 methylated H3 tail peptide leads to a dramatic increase in $K_m$ for peptide substrates (Supplementary Figure 3c). This increase in $K_m$ was particularly high for 10mer peptides, both in

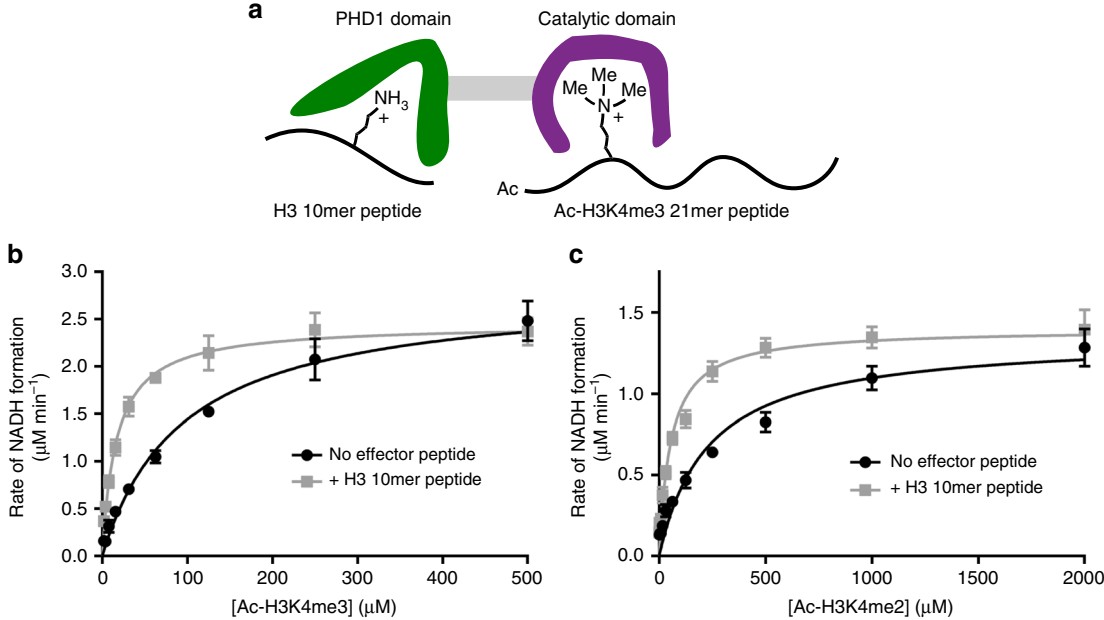

**Fig. 2** The catalytic efficiency of KDM5A is enhanced upon ligand binding to PHD1. **a** Schematic representation of the discrete occupancy activity assay in which an N-terminally acetylated 21mer peptide substrate (H3 A1-A21) and a 10-mer PHD1 ligand peptide (H3 A1-S10) are used. **b** Michaelis–Menten plot of KDM5A-catalyzed demethylation of Ac-H3K4me3 21mer peptide in the presence and absence of a saturating concentration (38 μM) of unmodified H3 10mer effector peptide. **c** Kinetic parameters of demethylation of the Ac-H3K4me2 21mer substrate as a function of the presence of the effector peptide. Both (**b**) and (**c**) are fitted to the Michaelis–Menten equation. Kinetic parameters are shown in Table 2. Errors ($n \geq 3$) represent s.e.m.

**Table 2 Kinetic parameters of KDM5A$_{1-797}$ reacting with various substrates with and without effector peptide**

| | Ac-H3K4me3 21mer | | Ac-H3K4me2 21mer | |
|---|---|---|---|---|
| | No effector peptide | +H3 10mer peptide | No effector peptide | +H3 10mer peptide |
| $k_{cat}$ (min$^{-1}$) | 2.82 ± 0.06 | 2.45 ± 0.04 | 1.35 ± 0.11 | 1.40 ± 0.06 |
| $K_m$ (μM) | 95.5 ± 7.6 | 16.8 ± 1.1 | 229 ± 60 | 54.7 ± 9.9 |
| $k_{cat}/K_m$ (mM$^{-1}$ min$^{-1}$) | 29.4 ± 3.3 | 146 ± 12 | 5.88 ± 2.04 | 25.6 ± 5.7 |

the di- and trimethylated state ($K_m > 750$ μM; Supplementary Figure 3b, c). Combined with our observation that H3 10mer peptides bind efficiently to the PHD1 domain (Fig. 1, Table 1), we selected the H3 10mer peptide (Ala1-Ser10) as ideal effector peptide for our experiments. Based on this information, a discrete occupancy activity assay was developed that utilizes a N-terminal H3 tail 10mer peptide as PHD1 effector ligand (H3 Ala1-Ser10) and a N-terminally acetylated 21mer H3K4me3 peptide as the catalytic domain substrate (Ac-H3K4me3 Ala1-Ala21) (Fig. 2).

We measured the Michaelis–Menten kinetic parameters for substrate demethylation in the presence and absence of the PHD1 domain ligand. The $K_m$ and $k_{cat}$ values were determined by measuring the initial rate of demethylation by KDM5A using the formaldehyde release assay at varying concentrations of acetylated substrate. With the Ac-H3K4me3 21mer peptide alone, we observe a $K_m$ value of $95.5 \pm 7.6$ μM and a $k_{cat}$ of $2.82 \pm 0.06$ min$^{-1}$ (Fig. 2b, Table 2). However, in the presence of the H3 10mer effector peptide at a concentration that allows for the saturation of the PHD1 domain with the effector (38 μM, $20 \times K_d$) the $K_m$ is reduced to $16.8 \pm 1.1$ μM while $k_{cat}$ remains similar (Fig. 2b, Table 2). This approximate fivefold reduction in $K_m$ is also observed when the Ac-H3K4me2 21mer peptide is used as a substrate (Fig. 2c, Table 2). Importantly, the presence of the effector peptide does not lead to a significant change in $K_m$ of α-KG co-substrate (Supplementary Figure 4). The reduction in $K_m$ for the peptide substrate alone suggests that effector peptide

binding to the PHD1 domain allosterically promotes binding of peptide substrates in KDM5A.

**Substrate binding is improved upon PHD1 occupation.** A decrease in $K_m$ upon effector peptide binding to the PHD1 domain suggests that there is improved substrate binding to the catalytic domain under these assay conditions. To further investigate this possibility the direct binding of an N-terminally acetylated substrate peptide was determined using fluorescence polarization assay. Binding assays were conducted in the presence and absence of the effector peptide and FP signal monitored over time (Supplementary Figure 5).

We observe that at 1 min the substrate binds with similar affinity to KMD5A, both in the absence ($K_d = 56.9 \pm 7.9$ μM) and presence ($K_d = 62.5 \pm 5.4$ μM) of H3 10mer peptide (Fig. 3). However, with no effector peptide, present the fluorescence polarization signal decreases over time. In contrast, when the H3 10mer peptide is present the signal is stable for at least 1 h. This striking difference between the two conditions suggests that the substrate peptide likely dissociates from the catalytic domain over time in the absence of the effector peptide, while the effector peptide occupied PHD1 domain stabilizes the interaction of substrate and the catalytic domain. Our data is consistent with a model where the demethylase is in a slow equilibrium between an open, substrate binding-competent state, and a closed state that does not bind the substrate. Binding of the effector peptide to the

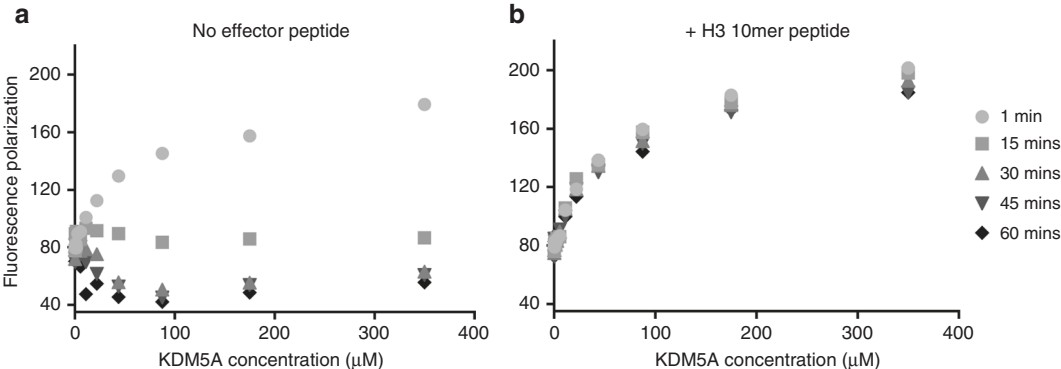

**Fig. 3** PHD1 occupation stabilizes substrate binding to the catalytic domain. Change in fluorescence polarization of C-terminally 5-carboxyfluorescein (5-FAM) conjugated Ac-H3K4me3 31mer peptide with varying concentrations of KDM5A$_{1-797}$, in the absence (**a**) or presence (**b**) of the H3 10mer effector peptide. Both assays contained 400 µM MnCl$_2$ and 400 µM N-oxalylglycine (N-OG). In panel (**b**), 350 µM H3 10mer effector peptide was used

PHD1 domain could shift the equilibrium between the two states toward the substrate binding-competent state (Supplementary Figure 6). Our observations that the substrate binding is stabilized in the presence of the PHD1 ligand could explain the decrease in $K_m$ that we observe when an effector peptide is present.

**Impact of H3 modifications on allostery.** The impact of post-translational modifications of H3 on binding of reader domains has been investigated thoroughly[11,12,40,47–51]. However, it is largely unknown how these modifications may regulate processes beyond effector-reader interactions, such as allostery.

To begin to investigate how modifications impact allostery in KMD5A, we performed competition-based FP assays to determine affinities of a panel of modified peptides (Fig. 4a). In analogy to our earlier findings the PHD1$_{V291-K347}$ construct[40], we observe that both methylation and mutation of H3R2 reduces binding affinities, emphasizing the importance of unmodified H3R2 in the PHD1 and H3 interaction. Phosphorylation of either the H3T3 or H3T6 residue greatly reduces the affinity, with H3T3 phosphorylation having a more pronounced effect. Methylation of H3R8 or H3K9 as well as acetylation of H3K9 has a modest or no effect on the affinity for the PHD1 domain. This agrees with the previous observation that the PHD1 domain predominantly interacts with the first five residues of the H3 peptide[40,44].

We investigated what impact modifications of the effector peptide have on demethylase activity using our discrete occupation activity assay. Specifically we determined the Michaelis–Menten parameters for demethylation of the Ac-H3K4me3 21mer substrate in the presence of saturating concentrations of various PHD1 ligand peptides (Fig. 4a). Similar to unmodified H3 (Fig. 3), we observed a significant decrease in the $K_m$ value of the substrate peptide for all modified effector peptides, with no significant to modest changes in the $k_{cat}$ value (Fig. 4a, b). The reduction in $K_m$ ranged from 3 to 6.5-fold, with the highest reduction in $K_m$ observed with the K9me3 and K9ac effector peptides, in addition to the unmodified H3 10mer peptide. This could be due to the lack of any H3 modifications that could interfere with optimal ligand engagement with the PHD1 domain in these effector peptides.

Together, our investigations of the impact of H3 modifications on PHD1 binding and KDM5A catalysis indicate a similar reduction in the $K_m$ of demethylation when PHD1 is saturated with effector peptide (Fig. 4b). However, modifications have a profound effect on the interaction of the PHD1 with H3 ligands, which would determine the concentrations of the PHD1 ligand required to saturate. Our findings indicate that the chromatin modifications context can impact the catalytic activity of KDM5A

through modulation of the degree to which PHD1 is bound by a ligand.

**PHD1 domain deletion affects substrate binding to KDM5A.** The observation that effector peptide binding to the PHD1 domain reduces the $K_m$ for demethylation raises questions about how the liganded PHD1 domain enables enhanced interaction of the demethylase with its substrate. In order to determine the effect of the PHD1 domain deletion on the kinetics of demethylation, a ΔPHD1 KDM5A construct was generated. While deletion constructs of KDM5 enzymes have been generated previously, these constructs lack both the ARID and PHD1 domains[14,16,35]. Our ΔPHD1 construct retains both the necessary JmjN and JmjC domains, as well as the ARID and ZF domains. The demethylase activity of the resulting ΔPHD1 construct toward the Ac-H3K4me3 21mer substrate was evaluated, and compared to the WT KDM5A$_{1-797}$ construct.

Despite similar $k_{cat}$ values of the two constructs, an approximate eightfold increase in $K_m$ is observed for ΔPHD1 KDM5A-catalyzed demethylation of acetylated substrate (Fig. 5, Table 3). These findings suggest that the binding of the substrate to the catalytic domain is impaired in the absence of PHD1. While we cannot exclude the possibility that the introduction of a GS linker may obstruct substrate binding, our observation that $k_{cat}$ is unaffected indicates that the linker does not inhibit catalysis. Our findings suggest that the PHD1 domain could restructure the catalytic domain to enhance substrate binding, or alternatively directly contribute to substrate binding.

**Assessing active site changes in the enhanced state.** Currently available structures of KDM5 enzymes lack the PHD1 domain, which is either disordered[15] or absent in the crystallized construct[16,35]. Without structural information, it is difficult to determine how effector peptide PHD1 binding-induced allosteric regulation is communicated between the PHD1 and catalytic domains.

In order to investigate the conformational mobility of KDM5A upon effector peptide binding to the PHD1 domain we performed HDX-MS analysis. HDX-MS measures the rate at which amide hydrogen atoms exchange with deuterium in the solvent and can report on regions of a protein that undergo changes in hydrogen bonding networks upon ligand binding. We applied HDX-MS to KDM5A$_{1-797}$ under conditions reflecting those of the discrete occupancy assay (Figs. 2 and 4). The conditions assayed were: KDM5A$_{1-797}$ bound to the PHD1-specific H3 10mer peptide; KDM5A$_{1-797}$ bound to the catalytic domain-specific Ac-H3K4me3 21mer substrate; and KDM5A$_{1-797}$ bound to both

**a**

|  | Modified peptide | PHD1 binding | Demethylase activity | |
|---|---|---|---|---|
|  |  | $K_d$ (µM) | $K_{cat}$ (µM) | $K_m$ (µM) |
| No effector | NA | NA | 2.82 ± 0.06 | 95.5 ± 7.6 |
| H3 | A R T K Q T A R K S | 0.996 ± 0.104 | 2.91 ± 0.05 | 16.8 ± 1.1 |
| H3R2me2a | A R T K Q T A R K S | 5.41 ± 0.49 | 1.66 ± 0.04 | 31.3 ± 3.2 |
| H3R2me2s | A R T K Q T A R K S | 5.31 ± 0.48 | 3.40 ± 0.05 | 25.2 ± 2.2 |
| H3R2A | A A T K Q T A R K S | 205 ± 73 | ND | ND |
| H3T3ph | A R T K Q T A R K S | NB | ND | ND |
| H3K4me1 | A R T K Q T A R K S | 3.49 ± 0.92 | 1.63 ± 0.03 | 26.7 ± 2.1 |
| H3K4me2 | A R T K Q T A R K S | 13.0 ± 1.7 | 2.66 ± 0.09 | 42.7 ± 5.7 |
| H3K4me3 | A R T K Q T A R K S | 34.0 ± 5.5 | ND | ND |
| H3K4A | A R T A Q T A R K S | 1.58 ± 0.19 | 3.20 ± 0.08 | 28.1 ± 3.3 |
| H3T6ph | A R T K Q T A R K S | 72.0 ± 10.3 | ND | ND |
| H3R8me2s | A R T K Q T A R K S | 2.01 ± 0.18 | 3.17 ± 0.06 | 25.2 ± 2.2 |
| H3R8me2a | A R T K Q T A R K S | 1.84 ± 0.17 | 3.19 ± 0.08 | 23.2 ± 2.4 |
| H3K9me3 | A R T K Q T A R K S | 0.74 ± 0.08 | 2.98 ± 0.05 | 18.6 ± 1.5 |
| H3K9ac | A R T K Q T A R K S | 1.01 ± 0.09 | 2.91 ± 0.04 | 14.6 ± 0.9 |

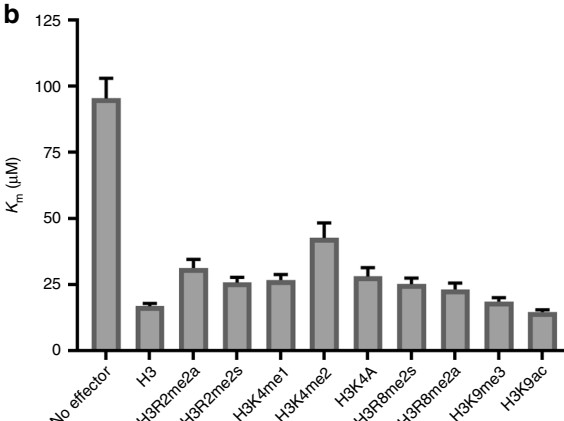

**Fig. 4** H3 modifications dictate demethylation activity enhancement in KDM5A. **a** Binding affinities ($K_d$) of various modified H3 10mer N-terminal peptides for PHD1 and the Michaelis–Menten parameters ($k_{cat}$ and $K_m$) for demethylation of Ac-H3K4me3 21mer substrate peptide in the presence of saturating concentrations of modified H3 10mer peptides. $K_d$ values were determined using a competition-based fluorescence polarization assay, and kinetic parameters for demethylation were measured by the formaldehyde release assay. *NB* no binding, *ND* not determined. **b** Graphical representation of the $K_m$ values in the presence of saturating concentrations of differently modified effector peptides. Error bars refer to the standard deviation. Errors ($n \geq 3$) represent s.e.m.

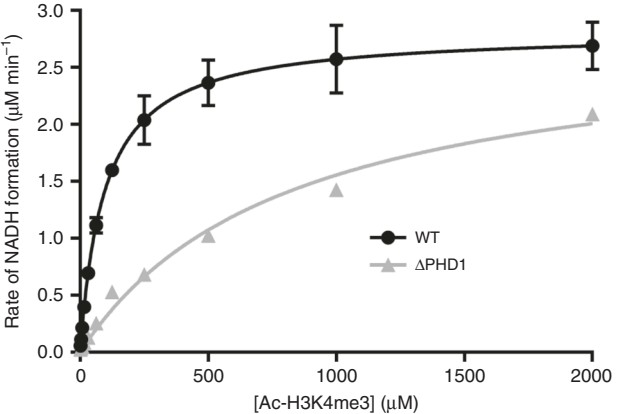

**Fig. 5** Deletion of the PHD1 domain impairs substrate binding by KDM5A. Michaelis–Menten plot for demethylation of Ac-H3K4me3 21mer substrate by KDM5A$_{1-797}$ and KDM5A$_{1-797}$ ΔPHD1 determined by the formaldehyde release assay where Ac-H3K4me3 21mer peptide was used as a substrate. Data are fitted to the Michaelis–Menten equation. Kinetic parameters are show in Table 3. Errors ($n \geq 3$) represent s.e.m.

**Table 3 Kinetic parameters of KDM5A$_{1-797}$ WT and ΔPHD1 constructs reacting with Ac-H3K4me3 21mer substrate**

|  | KDM5A$_{1-797}$ WT | KDM5A$_{1-797}$ ΔPHD1 |
| --- | --- | --- |
| $k_{cat}$ (min$^{-1}$) | 2.82 ± 0.08 | 2.83 ± 0.21 |
| $K_m$ (μM) | 95.5 ± 7.6 | 819 ± 133 |
| $k_{cat}/K_m$ (mM$^{-1}$ min$^{-1}$) | 29.4 ± 3.3 | 3.46 ± 0.82 |

the PHD1-specific H3 10mer peptide and catalytic domain-specific Ac-H3K4me3 21mer substrate.

For all conditions tested, >90% coverage of KDM5A$_{1-797}$ was obtained thereby enabling us to confidently assign changes in HDX kinetics to various protein domains. While the majority of peptides assigned to KDM5A$_{1-797}$ showed statistically insignificant changes (gray regions) in the levels of deuterium uptake, various regions of KDM5A$_{1-797}$ undergo differential deuterium uptake upon peptide binding (Figs. 6a, b, S7, S8). When only the PHD1-specific H3 10mer peptide is present, residues Gly300-Ala342 of the PHD1 domain showed between 2 and 8% reduced deuterium uptake (referred to as protection from deuterium exchange) (Fig. 6a, green regions; Supplementary Figures 7a and 8a). The observation that the majority of the PHD1 domain (Fig. 6c), rather than just the previously mapped peptide binding residues[40], undergoes a change suggests a broad stabilization of the PHD1 domain upon H3 peptide binding. Conversely, when only the catalytic domain-specific Ac-H3K4me3 21mer substrate peptide was tested no perturbations are observed in the protein (Supplementary Figure 7b), despite using saturating concentrations of the substrate (25-fold over protein concentration, ~5 × $K_m$). While it may be expected that residues in the catalytic domain would undergo protection from exchange with solvent upon substrate binding, this lack of perturbation may be due to weak binding of the acetylated substrate peptide, consistent with its high $K_m$ measured (Fig. 2b, Table 2). Alternatively, substrate interactions with the catalytic domain could be predominantly hydrophobic or hydrogen bonds from the peptide to KMD5A$_{1-797}$ could be replacing hydrogen bonds in apo KDM5A$_{1-797}$. Previous HDX studies have also shown no perturbation in hydrogen/deuterium exchange upon ligands binding[52]. However, when both the PHD1-specific H3 10mer peptide and the catalytic

domain-specific Ac-H3K4me3 21mer substrate peptide are present, two regions of KDM5A$_{1-797}$ undergo significant changes (Fig. 6b, Supplementary Figures 7c, 8c). In addition to the increased protection observed in the PHD1 domain (residues Gly300-Ala342), as in the presence of the H3 10mer peptide alone, a protein segment within the catalytic domain (residues Ser528-Leu545) experiences a statistically significant increase in deuterium uptake (3–4%) suggesting decreased protection (Fig. 6b, yellow regions). These observed perturbations map onto a helical loop region between the fourth and the fifth β-strand of the DSBH motif in the JmjC domain at the interface with the ARID domain (Fig. 6d). These findings reveal a conformation alteration in KDM5A$_{1-797}$, only present in the PHD1 effector bound higher activity state of KDM5A$_{1-797}$. Furthermore, the observation that helical loop region becomes more exposed only in the presence of both the effector and substrate peptide indicates a high degree of cooperativity between occupancy of these two sites. Such cooperativity helps explain the coupled energetic effects on catalysis seen in the context of substrate and effector binding (Fig. 2).

While currently available structural data does not allow for clear identification of the role of the helical loop region, we hypothesize that this region may be involved in binding of the methylated peptide substrate. This hypothesis comes from an overlay of structures of KDM5A and the structurally related *Arabidopsis thaliana* demethylase JMJ14 (53.2% sequence identity, 73.4% similarity between catalytic domains of KDM5A and JMJ14 with an RMSD of 0.613) in complex with a H3K4me3 peptide substrate[53]. Many residues of JMJ14, shown to interact with the H3K4me3 peptide substrate, are conserved in KDM5A[53] and comparison between the structures suggests a potential substrate binding groove in KDM5A (Supplementary Figure 9). The difference in length between the 21mer peptide substrate used in our experiments and the 7mer peptide substrate co-crystallized with JMJ14 makes it challenging to compare the residues that participate in binding of the C-terminal end of the peptide substrate. However, several residues conserved in between JMJ14 and KDM5A—Gln535, Leu536 and Val537—bind the H3K4me3 peptide in JMJ14 and are part of the helical loop region that we observe to undergo increased deuterium uptake when both PHD1 effector peptide and the substrate peptide are present. This region has reduced solvent protection only in the presence of both the substrate and effector peptides, and could be responsible for the stabilized substrate binding we have observed in direct binding experiments (Fig. 3).

In order to evaluate if any additional structural changes occur within KDM5A upon effector peptide binding to the PHD1 domain, lysine-directed crosslinking experiments were conducted (Supplementary Figure 10). Isotopically coded reagents were used to probe changes in crosslinking between unoccupied KDM5A and KDM5A with effector peptide bound. Lysine-lysine crosslinks were mostly distributed within the ARID domain and the disordered region linking ARID and PHD1 domains. In contrast, few crosslinks were found to JmjN, JmjC, PHD1, and zinc finger domains. Chemical crosslinking mass spectrometry relies on the presence of accessible, un-hydrogen bound lysine residues located within ionizable tryptic peptides, and is known to preferentially sample disordered protein regions while providing sparse coverage of beta sheets[54]. The quantitative comparison of crosslinks showed minor changes in crosslink occupancy and distribution when effector peptide is present (Supplementary Figure 10). Other than crosslinks between H3 effector peptide and KDM5A, few crosslinks were enriched in either condition more than twofold. Interactions between ARID and the PHD1-ARID linker are somewhat enhanced in the presence of effector (red-colored symbols), while ARID-ARID crosslinks and ARID-linker crosslink are enriched in the absence of effector (blue-colored symbols). However, the magnitude

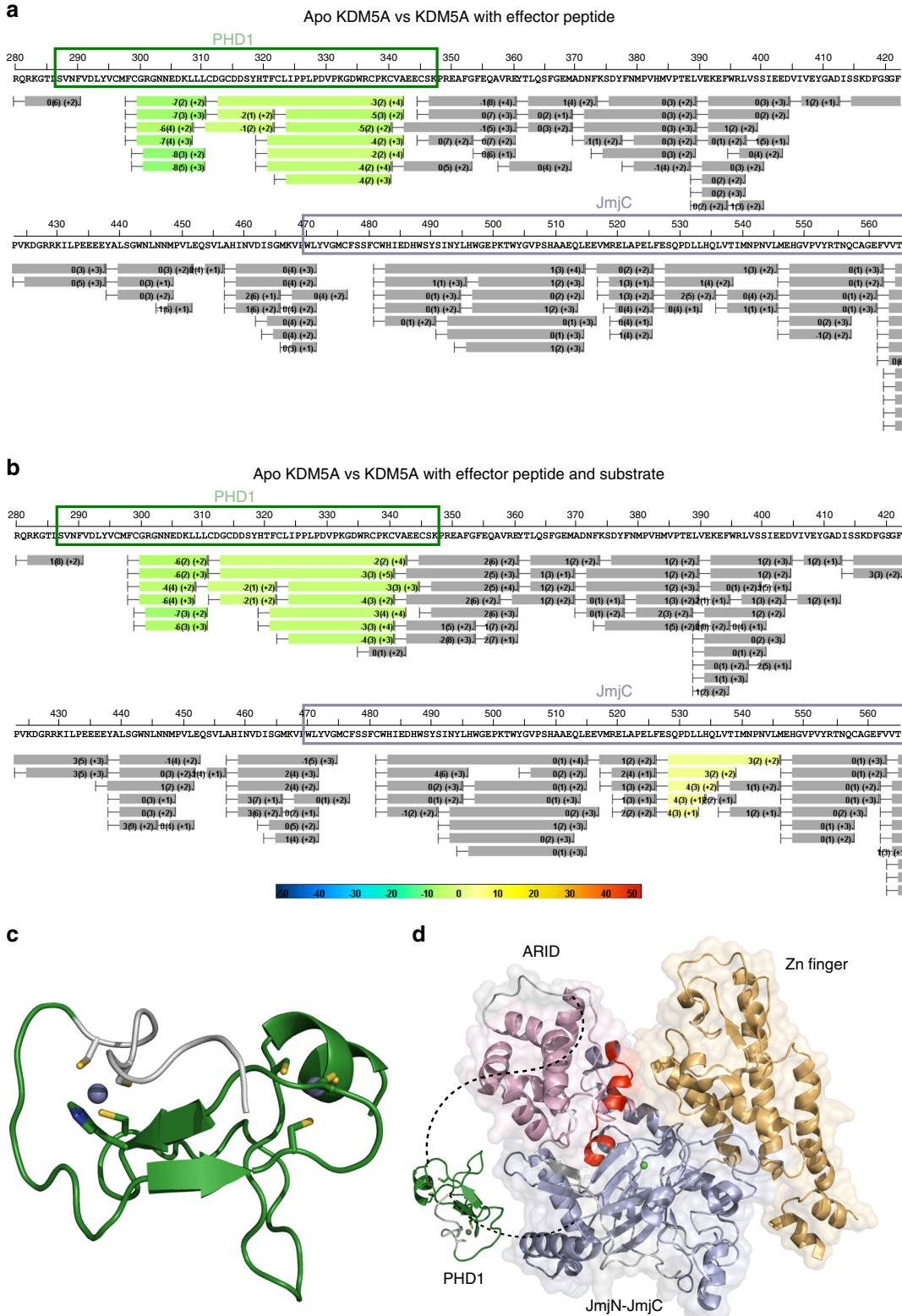

of these crosslink peak area changes relative to the variance due to measurement error, combined with a lack of structural information for the PHD1 and PHD1-ARID linker region, makes these data challenging to interpret.

## Discussion

Here we show that the allosteric stimulation of KDM5A activity, caused by binding of H3 N-terminal tail effector peptides to the

PHD1 domain, is due to a reduced $K_m$ value of peptide substrate demethylation. This was determined by using a novel in vitro activity assay that allows us to discretely occupy the two H3 binding sites in KDM5A. Furthermore, upon assessing the direct binding of peptide substrate to the demethylase we attribute this change to stabilization of the KDM5A-substrate complex only when the PHD1 domain is occupied. The enhancement in demethylation catalysis is largely independent of the type of modification present on the effector

**Fig. 6** HDX-MS detects changes in KDM5A$_{1-797}$ upon occupation of the PHD1 and catalytic domains. **a, b** Deuterium uptake plot of residues 280–570 of KDM5A, comparing **a** KDM5A·Fe(II)·N-OG (apo KDM5A) with KDM5A·Fe(II)·N-OG·H3 10mer peptide and **b** KDM5A·Fe(II)·N-OG (apo KDM5A) with KDM5A·Fe(II)·N-OG·H3 10mer·H3K4me3 21mer peptide. Deuterium uptake for each peptide is calculated as the average of %D uptake for the six time points (10, 30, 60, 300, 900, and 3600 s) and the difference in average %D values between the apo-KDM5A and KDM5A-peptide bound samples is shown as a heat map with a color code given at the bottom of the figure (warm colors for deprotection and cool colors for protection). Peptides are colored by the software automatically to display significant differences, determined either by a >5% difference (less or more protection) in average deuterium uptake between the two states, or by using the results of unpaired $t$-tests at each time point ($p$-value < 0.05 for any two time points or a $p$-value < 0.01 for any single time point). Peptides with nonsignificant changes between the two states are colored gray. The exchange at the first two residues for any given peptide is not colored. Each peptide bar in the heat map view displays the average Δ%D values, associated standard deviation, and the charge state. **c** Model of the PHD1 domain of KDM5A determined by NMR[40]. Residues that undergo a change in deuterium uptake are highlighted in green. Residues shown are those that coordinate the Zn atom, which is shown in purple. **d** Crystal structure of KDM5A$_{1-797}$ (PDB ID 5CEH)[15], in which the disordered PHD1 domain is represented by a dashed line and an image. The residues that experience a change in deuterium uptake are highlighted in red. The catalytic domain comprised of the JmjN and JmjC domains are shown in light blue, the ARID domain is shown in light pink and the zinc finger domain is shown in light orange. The Ni atom, which occupies the iron binding site in the crystal structure, is shown in lime green

peptide. However, it is strongly dependent on the occupancy of the PHD1 domain which directly correlates to the affinity of the effector peptide to the PHD1 domain. Therefore it is the affinity for the PHD1 domain that will dictate activity enhancement. Using HDX-MS, we identified a region in the catalytic domain that becomes more exposed only when both the PHD1 domain ligand and the substrate peptide are present. This structural change could be responsible for the stabilized substrate binding and the decrease in $K_m$ that we observe in demethylation assays when effector peptide is present.

Active regulation of the catalytic activity of KDM5A by H3 tail as an allosteric ligand has important implications. The observation that the allosteric stimulation is preserved with modified peptides, as long as the PHD1 domain is saturated, suggests that both the nature of the modification and the local concentration of the effector peptide can influence catalysis. Despite the low affinity of PHD1 for H3K4me3 marks[40], the local concentration of H3K4me3 marks at the sites of recruitment[20,55,56], can enable allosteric enhancement. Once generated, products H3K4me2, H3K4me1, and non-methylated H3 peptides would further enhance the catalytic efficiency of KDM5A due to their higher affinities for the PHD1 domain. Our previous evaluation of nucleosome demethylation suggests that the magnitude of stimulation in the context of chromatin substrate can be far greater than what is observed with the peptides in this study[40]. This positive feedback mechanism, where the reaction is stimulated by the product of the reaction, could contribute to rapid spreading of demethylation of H3K4me3-rich regions, which has been shown to occur in development, during maternal to zygotic transition[41–43] and in regulation of HOX gene expression[18,57]. However, rapid removal of these marks and the consequent repression of genes must be tightly regulated as dysregulation of boundary regions between active and silenced genes can disrupt cellular functions and lead to disease[58]. Our findings implicate PHD1 domain as a possible regulator of demethylation spread.

There are additional examples of allosteric stimulation of enzymes involved in epigenetic modifications[59]. For example, propagation of H3K27 methylation by the PRC2 complex is mediated by product binding to an EED domain within this complex[60]. Similar positive feedback regulation is observed in the Suv39h class of histone methyl transferases where H3K9 methylation is enhanced by the binding of the product of the reaction to a chromodomain within the methyltransferase[61–63]. Additional examples of allosteric regulation in epigenetic enzymes include acetyl transferases p300/CBP, Gcn5[59], DNA methyltransferase DNMT3A[64], and DNA demethylase Tet2[65]. Our studies have provided mechanistic rationale for allosteric regulatory role of the PHD1 domain in KDM5A, which expands the function of the PHD class of reader domain in regulation of chromatin methylation. PHD domain mediated allostery has also been shown to occur in the RAG-1/RAG-2 complex where

H3K4me3 binding by the RAG-2 PHD finger changes the conformation of RAG-1 leading to enhanced substrate binding and cleavage of DNA at recombination signal sequences[66,67].

Our investigation of catalysis in PHD1 deletion construct indicate that the PHD1 domain plays an important role either directly or indirectly, in organizing the catalytic domain for optimal substrate binding. We show that KDM5A retains its catalytic activity after deletion of the PHD1 domain, but that its substrate binding is impaired as reflected in a higher $K_m$ value. This finding is also mirrored in studies performed on a ΔAP KDM5B construct which shows a large increase in $K_m$, although the precise domain responsible could not be pinpointed[16]. Together, these data further highlight the importance of PHD1 in KDM5 enzymes, and parallel in vivo observations that this domain is necessary for activity of KDM5B[44,68] and its drosophila homolog Lid[56].

Our HDX-MS experiments have uncovered residue level dynamics information that cannot be deduced from currently available structures, thereby providing insights into the structural changes in KDM5A that are induced by effector peptide binding to the PHD1 domain[15,16,35]. The binding of the effector peptide to the PHD1 domain does not only induce changes in the peptide binding region that we previously mapped[40] but rather affects protection of backbone amide protons throughout the PHD1 domain. Additionally, only in the presence of both the substrate and the effector peptide a helical loop region between the fourth and the fifth β-strand of the DSBH motif in the catalytic domain becomes less protected from solvent. Overlaying the structures of KDM5A and plant homolog JMJ14, bound to a H3K4me3 peptide substrate, suggests that this helical loop region is involved in substrate binding (Supplementary Figure 9). These data could therefore suggest that the improved substrate binding is caused by a potential effector peptide binding-induced repositioning of the PHD1 domain that is coupled to the conformation of the helical loop region which orients itself to optimally bind the substrate.

This possibility is yet to be validated, given a lack of a co-crystal structure between any member of KDM5 family and its H3K4me3 substrate. However, our direct binding experiments lend support to this hypothesis and suggest that occupation of the PHD1 favors a conformation that is able to better bind substrate peptides. While our studies have determined regions of conformational change within KDM5A, additional studies are needed in order to further define the interactions of the PHD1 domain and the catalytic domain, as well as their interaction with chromatin. Our data cannot exclude the possibility that portions of the helical loop region, revealed by HDX-MS, may participate in domain communication within the demethylase.

Our findings contribute to the growing knowledge about allosteric regulation of epigenetic enzymes and highlight the complexity of mechanisms that enable context-dependent

regulation of chromatin marks. We hypothesize that the regulation of KDM5A activity by its PHD1 domain could be a mechanism of controlling demethylation spread. Recognition of H3 modifications by PHD1 could also be synergistic with other epigenetic regulatory enzymes. For example, H3R8 methylation supports optimal allosteric stimulation of KDM5A, but has a negative effect on activity of H3K4 methyltransferase MLL1. This could suggest a role where the chromatin context allows demethylation to occur while simultaneously lowering the opposing methylating activity of MLL1[69], keeping Lys4 in its unmethylated state. Further studies will be needed to address how allostery of KDM5A affects epigenetic regulation in a cellular context. Additionally, understanding the interface and conformational changes that occur in the higher affinity state of KDM5A could lead to future prospects of development of small molecule inhibitors of KDM5A that target its allosteric regulation.

## Methods

**Expression of recombinant WT and ΔPHD1 KDM5A.** WT KDM5A$_{1-797}$ was expressed in Sf21 cells following Invitrogen Bac-to-Bac baculovirus expression system protocol. KDM5A$_{1-797}$ was cloned into a pFASTBAC HTA vector. Purified bacmid was transfected in Sf21 cells. Approximately $0.8 \times 10^{-5}$ cells per well of a six-well dish were allowed to attach in 2 ml of SF-900 II SFM media containing 50 U ml$^{-1}$ penicillin and 50 µg ml$^{-1}$ streptomycin. While cells attached, 8 µl of Cellfectin II reagent (Invitrogen) in 100 µl of unsupplemented Grace's medium was mixed with ~2–5 µg of bacmid in 100 µl of unsupplemented Grace's medium and incubated for 15–30 min at 25 °C. Once cells were attached, media was removed and cells were washed with 2 ml of Grace's unsupplemented media. The bacmid DNA: Cellfectin mixture was then diluted to 1 ml with Grace's unsupplemented media and added to the well. Cells with bacmid:Cellfectin II mixture were incubated for 5 h at 27 °C. After 5 h of incubation, bacmid:Cellfectin mixture was removed and replaced with 2 ml of SF-900 II SFM 50 U ml$^{-1}$ penicillin and 50 µg ml$^{-1}$ streptomycin. Transfected cells were incubated 3–5 days or until signs of viral infection were observed. After transfection, the supernatant was spun down to remove the dead cells. The supernatant was then sterile filtered to obtain the P1 viral stock. To make P2, 20 ml of viral stock at ~2 × 10$^6$ cells per ml of sf21 was infected with 2 ml of P1 virus and incubated for 48–60 h. After 56 h, the cells were spun down and the supernatant was collected and sterile filtered to obtain P2 viral stock. Similarly, P3 viral stock from P2 viral stock was obtained. Generally, 1 l of Sf21 at 2 × 10$^6$ cells per ml was infected with ~40 ml of P3 virus for ~48–56 h. Cells were then collected and resuspended in the lysis buffer (25 mM HEPES pH 7.9, 350 mM NaCl, 5 mM KCl, 1.5 mM MgCl$_2$, 10 mM imidazole, aprotinin 2 µg ml$^{-1}$, leupeptin 3 µg ml$^{-1}$, pepstatin 3 µg ml$^{-1}$, and 1 mM PMSF). Cells were homogenized by emulsiflex. After lysis, the supernatant was recovered by centrifuging at 35k r.p.m. for 45 min, and incubated with cobalt resin equilibrated in lysis buffer for 1 h at 4 °C. After incubation the resin was washed with wash buffer (25 mM HEPES, pH 7.9, 350 mM NaCl, 0.5 mM MgCl$_2$, 10% glycerol, 10 mM imidazole, aprotinin 2 µg ml$^{-1}$, leupeptin 3 µg ml$^{-1}$, pepstatin 3 µg ml$^{-1}$, and 1 mM PMSF). His-KDM5A was eluted with elution buffer (25 mM HEPES pH 7.9, 100 mM NaCl, 0.5 mM MgCl$_2$, 10% glycerol, 100 mM imidazole, aprotinin 2 µg ml$^{-1}$, leupeptin 3 µg ml$^{-1}$, pepstatin 3 µg ml$^{-1}$, and 1 mM PMSF). The His-tag was removed by overnight incubation with TEV protease at 4 °C in the dialysis buffer (25 mM HEPES, pH 7.9, 100 mM NaCl, and 2 mM DTT). After cleavage, protein was further purified by size-exclusion chromatography using a S200 column. Purified KDM5A was eluted, aliquoted and stored at −80 °C in 40 mM HEPES, pH 7.9, 50 mM KCl.

The KDM5A$_{1-797}$ ΔPHD1 construct was generated after removing the residues Ser287-Lys347 and replacing them with a (GS)$_4$ linker before cloning into a pFASTBAC HTA vector. The protocol for expression and purification of the KDM5A$_{1-797}$ ΔPHD1 construct was the same as WT.

**Expression of GST-PHD1 (Ser287-Lys347).** All PHD1 constructs used (Val291-Lys247, S287-K347, and S287-K347 D292A) (Supplementary Figure 1) were cloned into a pET41a vector and expressed in BL21(DE3) *E. coli* cells. Expression and purification of all mutants followed the same protocol. Specifically, cells were induced with 0.4 mM IPTG and grown at 18 °C overnight before the pellet was collected. The cells were resuspended in lysis buffer (140 mM NaCl, 2.7 mM KCl, 10 mM Na$_2$HPO$_4$, 1.8 mM KH$_2$PO$_4$, 0.5 mM TCEP, 50 µM ZnCl$_2$, 1 mM phenylmethylsulfonyl fluoride (PMSF), pH 7.3), lysed by sonication and centrifuged. The supernatant was purified using Glutathione Sepharose 4B resin, washed with high salt buffer (50 mM HEPES, 700 mM KCl, 10% glycerol, 0.5 mM TCEP, 50 mM ZnCl$_2$, 1 mM PMSF, pH 8) and low salt buffer (50 mM HEPES, 150 mM KCl, 10% glycerol, 0.5 mM TCEP, 50 mM ZnCl$_2$, 1 mM PMSF, pH 8) and recovered by elution using low salt buffer with 30 mM glutathione. The sample was then concentrated and further purified by size-exclusion chromatography using a Hiload 26/60 Superdex 75 gel filtration column. Samples were eluted into 40 mM HEPES, 50 mM KCl, 0.5 mM TCEP, and 50 mM ZnCl$_2$ and flash-frozen.

**Fluorescence polarization binding assays.** The binding of GST-PHD1 to H3 10mer peptides was measured by either direct or competition-based fluorescence polarization (FP). For direct FP binding assay, 10 nM of C-terminal fluorescently labeled H3 peptide were incubated with varying concentrations of GST-PHD1. Data from direct FP measurements were fitted to equation 1:

$$FP_{obs} = \frac{FP_{max}[PHD1] + FP_{min}K_d}{K_d + [PHD1]}$$

For competition-based FP assays, 2 µM GST-PHD1 was incubated with 10 nM of C-terminal fluorescently labeled H3 peptide and different concentrations of unlabeled peptides were used as competitors. Data from competition-based FP assays were fitted to equation 2:

$$FP_{obs} = \frac{K_i(FP_{max}[PHD1] + FP_{min}K_d) + FP_{min}K_d[I]}{K_i(K_d + [PHD1]) + K_d[I]}$$

where FP$_{obs}$ is the observed FP, FP$_{max}$ is the maximum FP value, FP$_{min}$ is the minimum FP value, [PHD1] is concentration of PHD1, $K_d$ is the dissociation constant, $K_i$ is the inhibition constant referring to the competing peptide and [I] is the competing peptide concentration.

The binding of peptide substrates to KDM5A$_{1-797}$ was measured using the direct FP assay. A Ac-H3K4me3-fluorescein 31mer peptide was used, rather than the 21mer used in kinetic assays. This was in order to reduce interference of the C-terminal fluorescein in binding to KDM5A, since KDM5A is sensitive to changes in C-terminal tail of the H3 peptide (Supplementary Figure 3c). Four hundred micromolar MnCl$_2$ and N-OG were included to inhibit demethylase activity and 350 µM H3 10mer effector peptide was present when stated. A schematic showing the setup of measuring substrate binding is shown in Supplementary Figure 5.

**KDM5A demethylation assay.** A formaldehyde dehydrogenase (FDH)-coupled assay was used to monitor lysine 4 demethylation of histone H3 by following the production of formaldehyde during the KDM5A demethylation reaction. Reactions were performed in 50 mM HEPES, 50 mM KCl, 1 mM α-ketoglutarate, 50 µM Fe(NH$_4$)$_2$(SO$_4$)$_2$, 2 mM ascorbic acid, 2 mM NAD$^+$ and 0.05 U FDH (Sigma) at pH 7.5 at room temperature. Different concentrations of H3K4me3/2/1 peptides were incubated with 1 µM KDM5A. For assays measuring the stimulation by the effector peptide on the Michaelis–Menten kinetics, the effector peptide (differently modified H3 10mers Ala1-Ser10) was included in the reaction mix at 20× $K_d$ concentrations. There was an exception for the H3K4me2 10mer which was used at 10× $K_d$ due to its ability to act as a substrate at 20× $K_d$ (Supplementary Figure 3). Reactions were initiated by the addition of substrate and followed in 20 s intervals on a SpectraMax M5e (Molecular Devices) using 350 nm excitation and 460 nm emission wavelengths. An NADH standard curve was used to convert fluorescence to concentration of product formed. The initial 3 min were used to calculate initial velocities, which were plotted against substrate concentration. Michaelis–Menten parameters were determined using Graphpad Prism.

**Hydrogen/deuterium exchange-mass spectrometry (HDX-MS).** Solution-phase amide HDX experiments were carried out with a fully automated system (CTC HTS PAL, LEAP Technologies, Carrboro, NC; housed inside a 4 °C cabinet).

**Peptide identification for HDX-MS.** Peptides were identified using tandem MS (MS$^2$ or MS/MS) experiments performed with a LTQ linear ion trap mass spectrometer (LTQ Orbitrap XL with ETD, Thermo Fisher, San Jose, CA) over a 70 min gradient. Product ion spectra were acquired in a data-dependent mode and the five most abundant ions were selected for the product ion analysis per scan event. The MS/MS *.raw data files were converted to *.mgf files and then submitted to MASCOT ver2.3 (Matrix Science, London, UK) for peptide identification. The maximum number missed cleavage was set at 4 with the mass tolerance for precursor ions +/−0.6 Da and for fragment ions ± 8 ppm. Oxidation of methionine was selected for variable modification. Pepsin was used for digestion and no specific enzyme was selected in the MASCOT during the search. Peptides included in the peptide set used for HDX detection had a MASCOT score of 20 or greater. The MS/MS MASCOT search was also performed against a decoy (reverse) sequence and false positives were ruled out. The MS/MS spectra of all the peptide ions from the MASCOT search were further manually inspected and only the unique charged ions with the highest MASCOT score were used in estimating the sequence coverage and included in HDX peptide set.

**HDX-MS analysis.** For differential HDX experiments, 5 µl of 20 µM KDM5A$_{1-797}$ was diluted to 25 µl with D$_2$O-containing HDX buffer (40 mM HEPES, 50 mM KCl, pH 7.9) and incubated for 10, 30, 60, 900, and 3600 s. HDX conditions aimed to mimic the demethylation assay conditions as described in Fig. 2. Apo conditions were compared to the conditions with substrate and/or effector peptides. Apo refers to KDM5A with 50 µM FeSO$_4$ and 100 µM N-OG. The complex with substrate and/or effector peptides refers to KDM5A with 50 µM FeSO$_4$, 100 µM N-OG, and 500 µM Ac-H3K4me3 21mer substrate (25× [KDM5A]) with or without 100 µM H3 10mer effector peptide. Following on-exchange, unwanted forward or backward

exchange is minimized, and the protein is denatured by dilution to 50 µl with 0.1% TFA in 5 M urea and 50 mM TCEP (held at 4 °C, pH 2.5). Samples are then passed across an immobilized pepsin column (prepared in house) at 50 µl min$^{-1}$ (0.1% TFA, 15 °C), and the resulting peptides are trapped onto a $C_8$ trap cartridge (Thermo Fisher, Hypersil Gold). Peptides were then gradient eluted (5% $CH_3CN$ to 50% $CH_3CN$, 0.3% formic acid over 6 min, 4 °C) across a 1 mm × 50 mm $C_{18}$ analytical column (Hypersil Gold, Thermo Fisher) and electrosprayed directly into a high resolution orbitrap mass spectrometer (LTQ Orbitrap XL with ETD, Thermo Fisher). Percent deuterium exchange values for peptide isotopic envelopes at each time point were calculated and processed using HDX Workbench. Each HDX experiment was carried out in triplicate with a single preparation of each protein–ligand complex. The intensity weighted mean $m/z$ centroid value of each peptide envelope was calculated and subsequently converted into a percentage of deuterium incorporation. Statistical significance for the differential HDX data is determined by an unpaired $t$-test for each time point, a procedure that is integrated into the HDX Workbench software. Corrections for back-exchange were made on the basis of an estimated 70% deuterium recovery and accounting for 80% final deuterium concentration in the sample (1:5 dilution in $D_2O$ HDX buffer).

**Lysine crosslinking MS experiments**. Hundred micrograms of KDM5A$_{1-797}$ was crosslinked with 1 mM BS$^3$-H$_{12}$ for 1 h at 4 °C. KMD5A in the presence of the H3 effector peptide (40 µM) was crosslinked with 1 mM BS$^3$-D$_{12}$ (Creative Molecules) under the same conditions. After quenching with 50 mM Tris-base, the light and heavy crosslinked reaction mixtures were combined and precipitated with ice cold acetone. Crosslinked proteins were pelleted by spinning at $21,000 \times g$, the supernatant was removed, and the pellet was washed once with cold acetone. The precipitate was brought up in 8 M Urea, 10 mM TCEP, heated at 56 °C for 20 min, and then alkylated with 15 mM iodoacetamide (45 min at room temperature). The sample was diluted to 2 M Urea and digested overnight with 0.8 µg trypsin for 4 h at 37 °C. A second aliquot of trypsin was then added and digestion was allowed to proceed overnight. The digestion mixture was acidified to 1% TFA, desalted using a 100 µl OMIX C18 tip (Agilent), and evaporated to dryness. Crosslinked products were brought up in 10 µl of SEC buffer (70:30 $H_2O$:ACN with 0.1% TFA) and enriched by size-exclusion chromatography (Superdex Peptide, GE Healthcare Life Sciences). Hundred-microliter fractions eluting between 0.9 and 1.4 ml were dried, resuspended in 0.1% formic acid for MS analysis. The fractions starting at 0.9 ml and 1.3 ml were combined prior to evaporation to make four total SEC fractions.

LC-MS analysis was performed with a Q-Exactive Plus mass spectrometer (Thermo Scientific) coupled with a nanoelectrospray ion source (Easy-Spray, Thermo) and NanoAcquity UPLC system (Waters). Enriched fractions were separated on a 15 cm × 75 µm ID PepMap C18 column (Thermo) using a 70 min gradient from 2 to 23% solvent B (A: 0.1% formic acid in water, B: 0.1% formic acid in acetonitrile) followed by a 10 min gradient from 23 to 40%. Precursor MS scans were measured in the Orbitrap scanning from 350 to 1500 $m/z$ (mass resolution: 70,000). The ten most intense triply charged or higher precursors were isolated in the quadrupole (isolation window: 4 $m/z$), dissociated by HCD (normalized collision energy: 23), and the product ion spectra were measured in the Orbitrap (mass resolution: 17,500). A dynamic exclusion window of 15 s was applied and the automatic gain control targets were set to 3e$^6$ (precursor scan) and 5e$^4$ (product scan).

Peaklists were generated using Proteome Discoverer 2.2 (Thermo) and searched for crosslinked peptides with Protein Prospector 5.20.23. In total, 85 of the most intense peaks from each product ion spectrum were searched against a database containing the KDM5A construct and the H3 effector peptide concatenated with 10 randomized decoy versions of each of these sequences. Search parameters were as follows: mass tolerance of 10 ppm (precursor) and 25 ppm (product); fixed modifications of carbamidomethylation on cysteine; variable modifications of peptide N-terminal glutamine conversion to pyroglutamate, oxidation of methionine, and "dead-end" modification of lysine and the protein N-terminus by semi-hydrolyzed heavy and light BS$^3$; crosslinking reagents were BS$^3$-H$_{12}$ and BS$^3$-D$_{12}$; trypsin specificity was used with two missed cleavages and three variable modifications per peptide were allowed. Unique, crosslinked residue pairs were reported at a 0.5% FDR threshold, corresponding to a Score Difference cutoff of 10.

For quantitative analysis, precursor ion filtering in Skyline 4.1 was used to extract light:heavy crosslinked precursor ion signals. Skyline does not directly support crosslinking analysis, so the elemental composition of each crosslinked peptide species, precursor charge state, and retention time were imported as a small molecule transition list (generated in the R programming environment). Transitions were generated for every light or heavy crosslinked peptide species discovered in the Protein Prospector search and paired with its corresponding heavy or light counterpart. Theoretical elemental compositions were generated if the counterpart was not identified in the search results. Precursor ion transitions matching the first three isotopes were extracted across all four LC-MS fractions. Each extracted ion chromatogram was manually inspected and the start and end points were adjusted to ensure that the correct peaks were detected and that there were no interfering signals. The peak areas were re-imported into R and summarized at the level of crosslinked residues for each light and heavy crosslink. A weighted mean was used as the summarization function with weights being given by the peak areas (e.g., signals with larger peak area were weighted more heavily). Finally, log$_2$ ratios of the heavy (crosslinked in the presence of effector peptide) to light (absence of effector peptide) peak areas were determined.

**Reporting Summary**. Further information on experimental design is available in the Nature Research Reporting Summary linked to this article.

## Data availability
All relevant data supporting the key findings of this study are available within the article and its Supplementary Information files or from the corresponding author upon reasonable request. A reporting summary for this Article is available as a Supplementary Information file.

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

## Acknowledgements

We would like to thank Christian Freund, John Gross, Mark Kelly, Geeta Narlikar, Bruce Pascal and Fatima Ugur for their comments on the paper. This work is supported by National Institutes of Health (R01 GM114044 and R01 GM114044-03S1 to DGF and 1S10D016229 to AB) and the Dr. Miriam and Sheldon G. Adelson Medical Research Foundation (AB).

## Author contributions

J.E.L. and D.G.F. wrote the manuscript with input from V.D., M.T., A.B., and P.R.G. J.E. L., C.M.C., M.T., and V.D. designed and performed the experiments. I.O.T., D.G., and K. R. assisted in experimental design and initial experiments. All authors contributed to the interpretation and discussion of the results.

## Additional information

**Competing interests:** The authors declare no competing interests.

