## [Peer Review File · Nature Communications]

Reviewer #1 (Remarks to the Author):

The manuscript by Longbotham et al. addresses the question of the role(s) of one of the plant homeobox domains (PHD1) of the histone demethylase KDM5A. In essence the work shows blocking/binding of the substrate to the PHD1 enhances catalysis at the catalytic domain. The manuscript is essentially a follow up report from a previous report for the groups (Torres et al). At the biochemical level with peptide substrates the data is pretty good. However, it would be nice to have ITC or if not possible SPR/NMR binding data on the full length construct. As indicated below it's risky to correlate KM values with KD for such a complex reaction. More mutagenesis data on the full length protein would also be useful. What is missing in particular are results with full length histone substrates/nucleosomes. I think crucial to show these in order to make the claim the effector results are biologically relevant. History has shown it's dangerous to assume results with short peptides use always relevant with protein substrates.

Throughout the manuscript it is not always clear what results are relevant to the cellular context. I also think the manuscript is rather long. This is a one (albeit interesting) result paper; this would have come across more clearly without the multiple mini introductions.

Overall, my view is that the work could be of interest, but it would be crucial to show the effector idea is not an artefact of the 'in vitro' conditions. Hence, carrying out further kinetic analysis using nucleosomes and, ideally, cellular studies is important. It's crucial the biological relevance of the work is better validated.

Specific comments

Abstract:

The sentence beginning 'Using a discrete' needs work – the meaning of discrete occupancy needs defining.

It needs to be made clear that the results apply to artificial assays with isolated proteins.

P3 para 2 – the 'human' KDM5 subfamily Delete 'composite' (sense?) tone/claim statement on drug resistance – there is evidence...

P4 – it needs to make clear which experiments are with isolated components and which represent more biologically relevant situations, i.e. in cells.

Probably appropriate, at least, in the introductions to mention the regulation of other histone modifying enzymes, by PHD domains – this is probably best studied for the roles of the PHD domain in PHF8/KIA1718 catalysis – this situation is simpler than for KDM5A, but even in these relatively simple cases (1 x PHD domain adjacent to the JmjC domain), opposing effects are seen for PHF8 and KIA1718 on the effects of H3K4me3 binding to the PHD domain with regard to catalysis at H3Kme2; not sure if the effect on catalysis at other lysines are really understood. The key thing is that the regulation is complex.

P9 para – ‘a conserved carboxylate’ residue is used twice – please be precise and correlate with Zhang et al. figure 2. I don’t think ref 5 contains any crystallographic/NMR data so where is the discrepancy referred to?

P9 para 3 – quantify the increase in affinity.

Was the fold/thermodynamic stability of all the variants tested confirmed to be the same as wildtype by CD/NMR? I think more evidence, at least from ITC is needed to back up the electrostatic argument.

Note H3K4A still binds well to V291-K347 and binds less well to the PHD1 D29A variant.

Page 10 – opening sentence – make clear not in cells.

I really think ITC data would help ‘dissect contributions’. Should Michealis-Menten kinetics is surely difficult to interpret in such a complex reaction.

With regard to ‘exquisite control’ what are the limits of detection?

Was mutually exclusive binding of the reported peptides demonstrated by (e.g.) NMR titration experiments?

Surely it’s risky to assume K_M reflects binding when using a coupled assay in a complex reaction?

Do the K_M/K_{cat} values remain constant for the cofactors under the different conditions? (e.g. +/- effectors).

The use of the concentration assays with different modification is nice. But it would be nice to have K_D data for binding to the construct used for the demethylase activity tests.

(As an aside how were the concentrations of the peptides measured?)

Were the modified peptides tested with full length construct with a mutated Asp in PHD1?

The results in figure 4 suggest the importance of the PHD1 domain.

In the absence of KDM5 substrate complex structures use of H-D exchange is a good idea.

P18 – make clear if the work is on full length KDM5 or not (I think it was).

P20 – give % similarity/RMSD for human and Arabidopsis enzymes over relevant regions.

I’m not really clear about the conclusions from the H-DX work – how do they correlate with the crystallographically observed binding modes?

Discussion:

The results of a reduction in K_M by an effector peptide is interesting, but as with other studies on the roles of PHDs of the KDMs, there seems to be little biological evidence for the proposed allosteric regulation. This needs to be made clear.

Reviewer #2 (Remarks to the Author):

In this study by Longbotham et al, a role of PHD1 reader domain in allosteric regulation of KDM5A demethylation activity is established. This study presents a significant amount of biochemical and biophysical work to reveal the molecular basis underlying the allosteric enhancement in a quantitative manner. One major conclusion is that histone H3 binding by PHD1 stimulates enhanced binding of substrate (better K_m) to the demethylase domain of KDM5A. The authors further screened impact of differently modified H3 on the stimulatory activity of PHD1. HDX-MS experiment was then performed to confirm the conformational changes in PHD1 and regions of the catalytic domain responsible for substrate binding. Overall, this is an interesting study and paves the way for the development of small molecular allosteric modulators of KDM5A.

However, despite strong biochemical data, the molecular and structural insights disclosed by the current work are rather blurry and limited. Thus, for this manuscript to achieve the rigor required for Nature Communications, additional experiments are necessary to justify its publication.

1) A complex structure of PHD1 bound to unmodified H3 is necessary. It is unclear to this reviewer why D292 is required for high affinity binding. It is also strange that H3K4A displays a quite decent binding affinity (1.58 μM).

2) The information learned from HDX-MS is too blurry. It is unclear why PHD1-H3 interaction could promote H3K4me substrate peptide binding. It would be great if a complex structure of KDM5A(1-797) bound to substrate peptide could be determined.

3) If 2) is hard to achieve, at least cross-linking MS should be performed to identify the interaction surfaces between PHD1 and the catalytic domain of KDM5A.

4) The discussion is too lengthy and speculative. It should be kept as concise as possible.

5) It has been reported that ADD-H3 interaction could allosterically regulate the DNA methyltransferase activity of DNMT3A (Guo et al, Nature 517, 640–644, 2015). This work should be referenced and discussed.

Reviewer #3 (Remarks to the Author):

The authors explore allosteric communication and regulation of a chromatin reader domain (PHD1) on the catalytic activity of the histone demethylase KDM5A. Dissecting allosteric binding from catalytic activity appears to be daunting here, as the reader domain shares overlapping ligand affinity with the catalytic domain. One of the most significant advances of this work is to develop a different series of ligands/substrates to minimize overlap and dissect discrete functions of the PHD1 and catalytic domains. To that end, the authors test and characterize peptides modified to enhance functional orthogonality. Ultimately, this panel of ligand effectors and substrates are profiled to develop a model of regulatory allosteric communication between the reader and catalytic domain. To explore mechanisms of allosteric communication, the authors employ HDX-MS to map structural perturbations induced by PHD1 ligand occupancy.

The work is broadly interesting to the chromatin remodeling community as well as the much wider field of multi-domain sensor-coupled enzymes. The orthogonal ligand/substrates are an exciting development. The HDX-MS analysis is notably comprehensive, rigorous, and well-documented, especially regarding statistical testing and methodology.

Major point:

Given that occupancy of the PHD1 domain decreases K_m of the demethylase, the authors interpret the lower K_m as enhanced binding of the substrate at various points in the manuscript. This interpretation is reasonable, but the demethylase assay appears to be sensitive to only one of the two products (formaldehyde release rather than peptide release). In this case, the observed k_{cat} does not encompass both turnover and product release. Is it possible that the allosteric communication between PHD1 does not increase binding affinity for substrate, but rather, increases the rate of peptide product release? This seems especially relevant given that Ac-H3K4me2 is also a good substrate of the catalytic domain (Fig 2d).

Minor points:

The authors test two possible mechanisms by which PHD1 occupancy could influence demethylase activity by removing the PHD1 domain. This domain deletion is internal to the ORF, retaining the N-terminal ARID, ZF, and JmjN domains. To better highlight the need for this new construct, could the authors briefly add context explaining the desirability of the extra domains here? (i.e., is PHD1-JmjC vs. JmjC inadequate?). I bring this up because internal domain deletion adds protein misfolding risks, as the authors appropriately note.

The demethylation assay methods are largely relegated to Ref 51 (which is thorough), but inclusion of the enzyme concentration(s) used would help confirm that Michaelis-Menten assumptions are warranted here.

Second reference to Figure 2a seems to fit better with Figure 2b

Generally, tables of kinetic parameters often include more significant figures than appropriate, given the reported error values. (e.g., 141.4 +/- 28.3 is more properly reported as 140 +/- 30).

HDX-MS results described in the Results section are appropriately circumspect regarding interpretation of exchange rate perturbations, focusing on H-bonding and "protection." The discussion section reverts to an interpretation focused on "solvent accessibility." Given the myriad factors influencing exchange rate, "protection" and/or "H-bonding dynamics" would be more appropriate here.

Fig 5d: Would it be possible to include arrows (or some other annotation) indicating where along the backbone of the structural model (5CEH) the orderly electron density is lost and resumed corresponding to the missing PHD1 domain?

Here are a few possible typos too:

pg 11 "although activity I reduced relative to.."

pg 13 "...unknown how these modifications may regulatory processes..."

pg 14, "reduction in Km reduction?"

pg 17 "in the first model would have"

Response to Reviewers

We thank the reviewers for their thoughtful assessment of our manuscript. Their suggestions, and experiments that we carried out to address them over the past four months, have substantially improved the manuscript. During this time, the following key revisions have been addressed:

- We measured direct substrate binding;
- We experimentally assessed stability of various mutant proteins;
- We provide further evidence that our assay indeed allows for selective engagement of the PHD1 and the catalytic domain by the effector and substrate peptides, respectively;
- We carried out crosslinking experiments to investigate domain interactions;
- We measured kinetic parameters for α KG co-substrate as a function of the effector peptide.

Numerous additional comments have also been addressed. Our point-by-point response to the reviewers' comments (in bold fold) is provided below.

Reviewer #1 (Remarks to the Author):

1. The manuscript by Longbotham *et al.* addresses the question of the role(s) of one of the plant homeobox domains (PHD1) of the histone demethylase KDM5A. In essence the work shows blocking/binding of the substrate to the PHD1 enhances catalysis at the catalytic domain. The manuscript is essentially a follow up report from a previous report for the groups (Torres *et al.*). At the biochemical level with peptide substrates the data is pretty good. However, it would be nice to have ITC or if not possible SPR/NMR binding data on the full length construct. As indicated below it's risky to correlate K_M values with K_D for such a complex reaction.

We agree with the reviewer that correlating K_m values with the K_d of substrate binding is risky. We therefore measured the direct binding of the peptide substrate to the catalytic domain, with the PHD1 domain either occupied or unoccupied by effector peptide.

By monitoring substrate binding over time, we observe a striking difference between the two conditions. While an initial substrate binding occurs under both conditions (1 min time points, Figure 3), in absence of the effector peptide the substrate appears to dissociate over time (Figure 3a). However, when the effector peptide is present the substrate-bound state remains stable for over 1 h (Figure 3b). This difference is quite striking and suggests that effector peptide binding to the PHD1 domain stabilizes substrate binding, and that this improved substrate binding is responsible for the reduction in K_m that we observe in our activity assays when the PHD1 domain is occupied by an effector peptide.

An additional results section (pages 16-18), a figure with the new data (Figure 3), and a model of allosteric enhancement of KDM5A (Figure S6) have been added to the revised manuscript.

Using fluorescence polarization to measure binding has given us a reliable method to assess substrate affinity. The quantities of protein that FP uses are much lower than required for ITC, an important consideration because of low protein yields with insect cell expression. Additionally, we wish to point out that, to our knowledge, affinity of peptide substrate for the

catalytic domain of any histone demethylase has not been published, suggesting that this is a non-trivial task. We are aware that affinities of small molecules to the catalytic domain of histone demethylases have been determined by ITC. However, these molecules generally binding more tightly than our peptide substrate e.g Horton *et al*, Cell Chem Bio (2016), 769.

2. More mutagenesis data on the full length protein would also be useful.

The effect of mutations has been explored in our previous work (Torres *et al*/ Nat Comm 2015) where we show that a mutation that prevents effector peptide binding to the PHD1 abrogates allosteric stimulation. This data is consistent with a model that we present in this manuscript. While we are in agreement with reviewer 1 that more mutagenesis work would be information, choosing an additional site or sites to mutate would be difficult and potentially difficult to interpret with the current structural information available. Additionally, obtaining KDM5A mutants requires insect cell expression, which is highly time consuming effort.

3. What is missing in particular are results with full length histone substrates/nucleosomes. I think crucial to show these in order to make the claim the effector results are biologically relevant. History has shown it's dangerous to assume results with short peptides use always relevant with protein substrates.

Unfortunately this experiment is not feasible. To perform kinetics analysis analogous to our peptide experiments, we would need to use conditions where nucleosomes concentration is ranging from sub-saturating to 10x K_m . Our past nucleosome saturation experiments have suggests a $K_m > 60$ μ M for non-acetylated, H3K4me3 nucleosomes (Torres *et al*/ Nat Comm 2015). However, to achieve orthogonality of sites engagement, that is a hallmark of our discrete occupancy assay, we would need to use N-terminally acetylated H3K4me3 nucleosomes, which are likely to have higher K_m than non-acetylated nucleosomes, since there is a 10-fold increase in K_m due to N-terminal acetylation of the peptide substrate. Even without accounting for this probable K_m increase, we would need >600 μ M nucleosomes for saturation.

Our calculation suggests that, using our most sensitive demethylation assay in order to minimize needed substrate, we would need 0.12 μ mol nucleosomes to obtain a single Michaelis-Menten curve (we imagine needing at least 4 of these, two biological replicates with and without the effector peptide, therefore 0.48 μ mol). In our lab, typical nucleosome reconstitution is done on 1-2 nmol scale, a scale compatible with obtaining and purifying sufficient amounts of modified histone via native chemical ligation (Torres *et al*/ Nat Comm 2015). This scale is at least 60 smaller than what we need for a single experiment. We also wish to point out that our preparatory scale for nucleosome assembly is even larger than that used by others (Wojcik *et al*/ Nat Commun. 2018 Apr 11;9(1):1394).

Alternatively, although not ideal as it would not report on K_m change that we are testing, we could attempt sub-saturating single turnover assay with nucleosomes as limiting reagents, similar to what we have done before (Torres *et al* Nat Commun 2015). However, based on our data (Torres, SI figure 9), as we are approaching concentrations of the enzyme that would allow saturation (>100 μ M enzyme), the reaction would become too fast to monitor by available

assays. An additional drawback of this assay is that we would need enzyme concentration to be in hundreds of micromolar range, which is difficult to achieve with insect cell expression.

We do appreciate reviewer 1's remark, but due to the reasons explained above, these experiments are not feasible.

4. Throughout the manuscript it is not always clear what results are relevant to the cellular context.

We have revised the manuscript in several places to clearly state that we are using *in vitro* assays with purified KDM5A throughout the manuscript.

“Using purified KDM5A we demonstrate that ...” – Introduction last paragraph (page 5)

“enhances the rate of demethylation by KDM5A *in vitro*” – Results, second section, first paragraph (page 14)

“Using a novel *in vitro* activity assay” - Added to paragraph 1 of discussion (page 28 (p 29 in highlighted changes version))

5. I also think the manuscript is rather long. This is a one (albeit interesting) result paper; this would have come across more clearly without the multiple mini introductions.

The text has been revised to substantially shorten the manuscript. The beginning of each section in the results section has been amended to make the text more concise. Several sections of the discussion have also been removed.

6. Overall, my view is that the work could be of interest, but it would be crucial to show the effector idea is not an artefact of the ‘in vitro’ conditions. Hence, carrying out further kinetic analysis using nucleosomes and, ideally, cellular studies is important. It's crucial the biological relevance of the work is better validated.

Please see our nucleosome comments above.

Unfortunately carrying out our assay in cells is not feasible. The hallmark of this manuscript is the use of orthogonal effector-substrate pairs, where effector peptide only engages the PHD1 domain, and the substrate peptide, due to its N-terminal acetylation, only engages the catalytic domain. Developing this system was the key to mechanistic dissection of allosteric stimulation in KDM5A. Endogenous cellular substrates unfortunately do not allow for recapitulation of these orthogonal engagement conditions.

7. Specific comments, Abstract: The sentence beginning ‘Using a discrete’ needs work – the meaning of discrete occupancy needs defining.

This has been amended. The text now reads: “Using a novel activity assay that enables selective engagement of the PHD1 and the catalytic domains by the effector and substrate peptides” (page 2).

8. It needs to be made clear that the results apply to artificial assays with isolated proteins.

As stated above (point number 4), we have amended the text to make it clear that our work is based on *in vitro* reconstitution, a common approach for mechanistic dissection of complex biological regulatory mechanisms.

9. P3 para 2 – the ‘human’ KDM5 subfamily Delete ‘composite’ (sense?) tone/claim statement on drug resistance – there is evidence...

“Composite” has been removed. – Introduction, paragraph 2, page 3.

Cancer drug resistance claim has been altered – “Additionally, there is evidence for overexpression” – Introduction, paragraph 2, page 3.

10. P4 – it needs to make clear which experiments are with isolated components and which represent more biologically relevant situations, i.e. in cells.

We have amended this in the text. See the response to point number 4.

11. Probably appropriate, at least, in the introductions to mention the regulation of other histone modifying enzymes, by PHD domains – this is probably best studied for the roles of the PHD domain in PHF8/KIA1718 catalysis – this situation is simpler than for KDM5A, but even in these relatively simple cases (1 x PHD domain adjacent to the JmjC domain), opposing effects are seen for PHF8 and KIA1718 on the effects of H3K4me3 binding to the PHD domain with regard to catalysis at H3Kme2; not sure if the effect on catalysis at other lysines are really understood. The key thing is that the regulation is complex.

Additional text has been added to the Introduction paragraph 3, page 4:

“PHD domains in demethylases have traditionally been associated with the recruitment of demethylases to chromatin. For example, PHD domain protein BHC80 is a component of LSD1 co-repressor complex that stabilizes the recruitment of LSD1 to chromatin. In addition, PHD domains can also regulate substrate specificity of demethylases, as in the case of PHF8 and KIAA1718.”

Appropriate references (refs “Lan, F. et al.. Nature 448, 718–722 (2007)” and “Horton, J. R. et al. *Nat. Struct. Mol. Biol.* **17**, 38–43 (2010)”) have been cited.

12. P9 para – ‘a conserved carboxylate’ residue is used twice – please be precise and correlate with Zhang et al. figure 2. I don’t think ref 5 contains any crystallographic/NMR data so where is the discrepancy referred to?

The discrepancy between PHD1_{KDM5A} and PHD1_{KDM5B} refers to the lack of any NMR evidence that the conserved Asp residue in KDM5A (D292) is involved in binding, which we believe is due to the construct that we previously used (Torres et al Nat Comm 2015). In the current

manuscript, we have shown that this residue is indeed important and that our construct behaves similarly to what has been shown for PHD1 in KDM5B by Zhang et al.

The text in Results section has been revised to make this clearer (page 11-13 of the revised manuscript).

13. P9 para 3 – quantify the increase in affinity.

The text has been updated with reference to differences in affinity between different proteins and ligands. E.g “shows a 2.3-fold preference for WT H3”; “similar 2.5-fold discrimination of WT H3” – Results, section 1, paragraph 3, page 12.

14. Was the fold/thermodynamic stability of all the variants tested confirmed to be the same as wildtype by CD/NMR? I think more evidence, at least from ITC is needed to back up the electrostatic argument. Note H3K4A still binds well to V291-K347 and binds less well to the PHD1 D29A variant.

We have assessed the stability experimentally to find that both the PHD1 WT and D292A variants have similar CD spectra, suggesting that secondary structure is not affected by the mutation (Figure S2). Additionally, the thermal unfolding of both constructs is similar (Figure S2).

Furthermore, the text of this section has been revised to make our conclusions with regard to the role of D292 in effector peptide binding clearer (page 11-13). The key point is that while this residue contributes to H3 binding, it is not necessary for it, consistent with the previous work by us and others (Zhang et al Protein Cell 2014; Torres et al Nat Comm 2015; Klein Cell Rep 2014).

15. Page 10 – opening sentence – make clear not in cells.

This has been amended, as stated above (please see points 4, 8 and 10).

“Page 10” (now page 13 (page 14 in the highlighted changes version)) text now reads: “of demethylation by KDM5A *in vitro*⁵⁵”

16. I really think ITC data would help ‘dissect contributions’. Should Michealis-Menten kinetics is surely difficult to interpret in such a complex reaction.

We have now measured binding of substrate to the catalytic domain by fluorescence polarization. This has been discussed in more detail in response to the point 1 above.

17. With regard to ‘exquisite control’ what are the limits of detection?

Our data shows that the concentrations of acetylated substrate peptide used in this assay (2 mM maximum for Ac-H3K4me2 and 0.5 mM maximum for Ac-H3K4me3) would bind very

weakly to the PHD1 domain (Figure S2a). Additionally, our data indicates that shorter peptides are poor substrates for KDM5A ($K_m = 770 \mu\text{M}$ for H3K4me3 10mer, Figure S3). The concentrations of effector peptide used in this assay are substantially below this ($38 \mu\text{M}$ for H3 10mer peptide) and therefore would not interact with the catalytic domain.

18. Was mutually exclusive binding of the reported peptides demonstrated by (e.g.) NMR titration experiments?

Using fluorescence polarization we experimentally demonstrated that that N-terminally acetylated peptides has a very weak affinity for the PHD1 domain (approximate K_d of 10 mM , Figure S2a). This is detailed in the text (page 14).

19. Surely it's risky to assume K_M reflects binding when using a coupled assay in a complex reaction?

In the revised manuscript we have measured substrate binding directly, as described in point 1 above.

20. Do the K_M/K_{cat} values remain constant for the cofactors under the different conditions? (e.g. +/- effectors).

To address this concern, we have measured the Michaelis-Menten kinetics of co-substrate α -KG in the presence and absence of effector peptide (**Figure S4**). In contrast to what we see with the substrate peptides, we show that upon addition of the effector peptide no significant change in K_m or k_{cat} is observed. By showing that the occupation of the PHD1 domain only has a significant effect on peptide substrate binding rather than the other reaction components, this data further supports our model that allosteric enhancement is due to improved binding of the peptide substrate.

21. The use of the concentration assays with different modification is nice. But it would be nice to have K_D data for binding to the construct used for the demethylase activity tests. (As an aside how were the concentrations of the peptides measured?)

We have done this previously (Torres et al, Nat Comm 2015, Figure 3A). Our findings indicate that binding of the effector to the PHD1 is preserved in the context of the full length demethylase.

Peptide solutions were prepared by dissolving carefully weighed lyophilized peptide ($>1 \text{ mg}$) in water, and making serial dilutions.

22. Were the modified peptides tested with full length construct with a mutated Asp in PHD1?

This exact experiment was not carried out, however we have done a highly similar experiment with a Trp mutant that abrogates effector peptide binding (Torres et al Nat Comm 2015), which showed, as expected, that allosteric communication is abrogated (see response to point 2 above). We have not done the experiment with Asp mutant since this experiment will require a lengthy insect cell expression of the new mutant protein.

23. The results in figure 4 suggest the importance of the PHD1 domain. In the absence of KDM5 substrate complex structures use of H-D exchange is a good idea. P18 – make clear if the work is on full length KDM5 or not (I think it was).

KDM5A in this section is now referred to as KDM5A₁₋₇₉₇.

24. P20 – give % similarity/RMSD for human and Arabidopsis enzymes over relevant regions. I'm not really clear about the conclusions from the H-DX work – how do they correlate with the crystallographically observed binding modes?

The similarities between the catalytic domains of JMJ14 and KDM5A were calculated to be 53.2% sequence identity, 73.4% similarity and RMSD between JmjC domains was 0.613.

Using HDX, we have identified a region of the catalytic domain that is altered upon formation of the allosterically enhanced enzyme (Figure 6). We wish to point out that currently there is no crystal structure of a KDM5 enzyme bound to a peptide substrate. Through comparisons with JMJ14 we are suggesting that this region is involved in substrate peptide binding (Figure S9). The conservation of peptide substrate binding residues between these two enzymes suggests a similar substrate binding mode.

25. Discussion: The results of a reduction in KM by an effector peptide is interesting, but as with other studies on the roles of PHDs of the KDMs, there seems to be little biological evidence for the proposed allosteric regulation. This needs to be made clear.

We have made additions to make it clear that all experiments are conducted *in vitro*. We also make it clear that there is future work to be done to understand allostery of KDM5A in a biological setting. The following text revisions were made to address this concern:

Added to paragraph 1 of discussion – “Using a novel *in vitro* activity assay”

“Further studies will be needed to address how allostery of KDM5A affects epigenetic regulation in a cellular context” – added to discussion, final paragraph, page 31 (page 34 in highlighted changes version).

Reviewer #2 (Remarks to the Author):

In this study by Longbotham et al, a role of PHD1 reader domain in allosteric regulation of KDM5A demethylation activity is established. This study presents a significant amount of biochemical and biophysical work to reveal the molecular basis underlying the allosteric enhancement in a quantitative manner. One major conclusion is that histone H3 binding by PHD1 stimulates enhanced binding of substrate (better K_m) to the demethylase domain of KDM5A. The authors further screened impact of differently modified H3 on the stimulatory activity of PHD1. HDX-MS experiment was then performed to confirm the conformational changes in PHD1 and regions of the catalytic domain responsible for substrate binding. Overall, this is an interesting study and paves the way for the development of small molecular allosteric modulators of KDM5A.

However, despite strong biochemical data, the molecular and structural insights disclosed by the current work are rather blurry and limited. Thus, for this manuscript to achieve the rigor required for Nature Communications, additional experiments are necessary to justify its publication.

1) A complex structure of PHD1 bound to unmodified H3 is necessary. It is unclear to this reviewer the why D292 is required for high affinity binding. It is also strange that H3K4A displays a quite decent binding affinity (1.58 μM).

We respectfully disagree with the reviewer that solving structure of the PHD1-ligand complex is necessary. The complex structure has been solved in highly related system that we cite, KDM5B's PHD1 (Klein Cell Rep 2014 and Zhang Protein Cell 2014). The two PHD1 domains share very high sequence identity (82% identity, and 95% similarity). Furthermore, mutation of the analogous Asp in this highly related PHD domain decreases the affinity for the H3 peptide by approximately 6-fold, which is similar to 7.6-fold change that we see in our system. Assays examining binding of PHD1_{KDM5B} to H3K4A peptide showed a similar fold-change in affinity to that described in our current manuscript. We have revised the text to provide a clearer summary of our findings with both D292A and H3K4A mutants (see page 11-13). Given the similarity in sequence and binding affinities, we believe that obtaining another structure of PHD1 domain would have limited value.

Additionally, we wish to point out that the focus of this submission is not the PHD1 domain-effector peptide interaction, but rather allosteric communication between the PHD1 and the catalytic domain.

2) The information learned from HDX-MS is too blurry. It is unclear why PHD1-H3 interaction could promote H3K4me substrate peptide binding. It would be great if a complex structure of KDM5A(1-797) bound to substrate peptide could be determined.

3) If 2) is hard to achieve, at least cross-linking MS should be performed to identify the interaction surfaces between PHD1 and the catalytic domain of KDM5A.

As the reviewer themselves suggests, obtaining a structure of KDM5A (1-797) bound to the substrate peptide is a very challenging task and studies solely dedicated to structural analysis of this complex enzyme have not yielded a crystal structure with the substrate nor the PHD1 domain (Vinogradova et al, Nat Chem Biol 2016).

We have taken reviewer's advice and carried out crosslinking mass spectrometry experiments. Lysine crosslinking experiments were conducted in order to identify any changes in domain interactions as a function of the effector peptide (**Figure S10**). Specifically, we compared differences in crosslinked residues between KDM5A₁₋₇₉₇ in the absence and presence of effector peptide bound to the PHD1 domain. Our findings are summarized on pages 27-28 (pages 28-29 of highlighted changes version) of the revised manuscript, as well as in **Figure S10a**. Although very few crosslinks change significantly under the assay conditions, suggesting lack of large scale conformational changes, it is interesting the crosslinks that are changing are located in the ARID domain, which is adjacent to the PHD1, and the ARID-PHD1 linker region.

Unfortunately, as we discuss in the manuscript, there is currently no structural information on the ARID-PHD1 linker region and the PHD1 domain in the context of KDM5A. This limitation makes interpretation of crosslinks difficult, as the majority of significantly altered crosslinks were between regions for which structural information is lacking.

4) The discussion is too lengthy and speculative. It should be kept as concise as possible.

Several sections of the discussion have been removed. These are indicated in the track changes of the updated manuscript.

5) It has been reported that ADD-H3 interaction could allosterically regulate the DNA methyltransferase activity of DNMT3A (Guo et al, Nature 517, 640–644, 2015). This work should be referenced and discussed.

This reference has been added to the text and discussed.

"Additional examples of allosteric regulation in epigenetic enzymes include acetyl transferases p300/CBP, Gcn5; DNA methyltransferase DNMT3A and DNA demethylase Tet2." (page 28/29 (page 32 of highlighted changes version))

Reviewer #3 (Remarks to the Author):

The authors explore allosteric communication and regulation of a chromatin reader domain (PHD1) on the catalytic activity of the histone demethylase KDM5A. Dissecting allosteric binding from catalytic activity appears to be daunting here, as the reader domain shares overlapping ligand affinity with the catalytic domain. One of the most significant advances of this work is to develop a different series of ligands/substrates to minimize overlap and dissect discrete functions of the PHD1 and catalytic domains. To that end, the authors test and characterize peptides modified to enhance functional orthogonality. Ultimately, this panel of ligand effectors and substrates are profiled to develop a model of regulatory allosteric communication between the reader and catalytic domain. To explore mechanisms of allosteric communication, the authors employ HDX-MS to map structural perturbations induced by PHD1 ligand occupancy.

The work is broadly interesting to the chromatin remodeling community as well as the much wider field of multi-domain sensor-coupled enzymes. The orthogonal ligand/substrates are an exciting development. The HDX-MS analysis is notably comprehensive, rigorous, and well-documented, especially regarding statistical testing and methodology.

Major point:

Given that occupancy of the PHD1 domain decreases K_m of the demethylase, the authors interpret the lower K_m as enhanced binding of the substrate at various points in the manuscript. This interpretation is reasonable, but the demethylase assay appears to be sensitive to only one of the two products (formaldehyde release rather than peptide release). In this case, the observed k_{cat} does not encompass both turnover and product release. Is it possible that the allosteric communication between PHD1 does not increase binding affinity for substrate, but rather, increases the rate of peptide product release? This seems especially relevant given that Ac-H3K4me2 is also a good substrate of the catalytic domain (Fig 2d).

We agree with the reviewer and have addressed this criticism experimentally, as described in the response to review 1 (point 1).

Additionally, Michealis-Menten measurements of co-substrate α -KG show no change in K_m and k_{cat} with effector peptide binding (please see our response to reviewer 1, point #20). This also supports our conclusion that PHD1 domain occupation affects substrate binding and that other reaction parameters assessed are unaffected.

Minor points:

The authors test two possible mechanisms by which PHD1 occupancy could influence demethylase activity by removing the PHD1 domain. This domain deletion is internal to the ORF, retaining the N-terminal ARID, ZF, and JmjN domains. To better highlight the need for this new construct, could the authors briefly add context explaining the desirability of the extra domains here? (i.e., is PHD1-JmjC vs. JmjC inadequate?). I bring

this up because internal domain deletion adds protein misfolding risks, as the authors appropriately note.

This paragraph has been rewritten (page 21 of the revised manuscript) to make it clearer that we intended to look at the effects of deleting only the allosteric regulatory domain (PHD1) rather than any of domains that are essential for catalysis (JmjC, JmjN and ZF). To further minimize changes to the protein, we have preserved the ARID domain in our construct.

The demethylation assay methods are largely relegated to Ref 51 (which is thorough), but inclusion of the enzyme concentration(s) used would help confirm that Michaelis-Menten assumptions are warranted here.

The enzyme concentration of KDM5A that was used in the assays (1 μ M) has been included in the methods section (page 7 of the revised manuscript).

Second reference to Figure 2a seems to fit better with Figure 2b

This has been amended; this text now refers to Figure 2 (page 15 of the revised manuscript).

Generally, tables of kinetic parameters often include more significant figures than appropriate, given the reported error values. (e.g., 141.4 +/- 28.3 is more properly reported as 140 +/- 30).

Figures 1, 2, 4 have been altered to give at most 3 significant figures.

HDX-MS results described in the Results section are appropriately circumspect regarding interpretation of exchange rate perturbations, focusing on H-bonding and “protection.” The discussion section reverts to an interpretation focused on “solvent accessibility.” Given the myriad factors influencing exchange rate, “protection” and/or “H-bonding dynamics” would be more appropriate here.

The text has been amended to address these concerns. It now reads:

“mapped but rather affects protection of backbone amide” –(page 30 (p 33 in the highlighted changes version))

“that helical-loop region becomes more exposed only” (page 24 (page 25 in the highlighted changes version))

“region has reduced solvent protection” (page 25 (page 26 in the highlighted changes version))

“becomes more protected from solvent upon...” (page 30 (page 33 in the highlighted changes version))

Fig 5d: Would it be possible to include arrows (or some other annotation) indicating where along the backbone of the structural model (5CEH) the orderly electron density is lost and resumed corresponding to the missing PHD1 domain?

We have revised that figure (Figure 6d in the amended manuscript) to insert dashed lines and a model of PHD1 (page 26 of the revised manuscript (page 27 in the highlighted changes version)).

Here are a few possible typos too:

pg 11 “although activity I reduced relative to..”

pg 13 “...unknown how these modifications may regulatory processes...”

pg 14, “reduction in Km reduction?”

pg 17 “in the first model would have”

All have been amended or deleted in the text.

Now on page 14, 18, 18 and deleted, respectively.

Reviewer #1 (Remarks to the Author):

Overall, I think the authors have addressed the major scientific/technical concerns raised by myself by the application in the fluorescence polarisation based assay; although this is not a substitute for ITC/SPR type analyses it does improve the manuscript. The question of the in vivo relevance of the results still applies. At least from a technical perspective the manuscript is acceptable, though I think it remains important to keep stressing that the in vivo relevance of the work is yet to be investigated. For example in the second sentence of the abstract make it clear this evidence arises from work with purified proteins. It also needs to be made clear that the differences observed were relatively small.

Minor points:

L63 - 'human KDM5 subfamily'.

L81 – 104 (C271 etc.) – make clear these are in vitro analyses.

L406 – post-translational

L384 – Are there errors on fig 3?

I do appreciate the efforts been made to make the manuscript more concise. (This is also a nice use of the HDX technique).

Reviewer #2 (Remarks to the Author):

"In the revision, the authors provided new results and a nice summary of the D292A and H3K4A mutants. The cross-linking MS results suggested the involvement of ARID domain in allosteric regulation. It is a pity that the molecular basis are still blurry due to lack of a real structure. However, the orthogonal ligand/substrate methodology is rather novel and important. The demonstration of an allosteric regulation mechanisms in the histone demethylase family (similar mechanism was previously reported for a DNA methyltransferase) is also interesting. Therefore, the paper can be accepted for publication in Nat Communications."

Reviewer #3 (Remarks to the Author):

I am satisfied with the revisions. The direct binding data and other supplementary experiments are a welcome addition.

REVIEWERS' COMMENTS:

Reviewer #1 (Remarks to the Author):

Overall, I think the authors have addressed the major scientific/technical concerns raised by myself by the application in the fluorescence polarisation based assay; although this is not a substitute for ITC/SPR type analyses it does improve the manuscript. The question of the in vivo relevance of the results still applies. At least from a technical perspective the manuscript is acceptable, though I think it remains important to keep stressing that the in vivo relevance of the work is yet to be investigated. For example in the second sentence of the abstract make it clear this evidence arises from work with purified proteins. It also needs to be made clear that the differences observed were relatively small.

Minor points:

L63 - 'human KDM5 subfamily'. Line 69

This has been added.

L81 – 104 (C271 etc.) – make clear these are in vitro analyses.

"In vitro" has been added twice to the introduction (line 86 and 90/91).

L406 – post-translational

Corrected.

L384 – Are there errors on fig 3?

No. This experiment has been conducted once due to the large concentrations of insect cell produced KDM5A required.

I do appreciate the efforts been made to make the manuscript more concise. (This is also a nice use of the HDX technique).

Reviewer #2 (Remarks to the Author):

"In the revision, the authors provided new results and a nice summary of the D292A and H3K4A mutants. The cross-linking MS results suggested the involvement of ARID domain in allosteric regulation. It is a pity that the molecular basis are still blurry due to lack of a real structure. However, the orthogonal ligand/substrate methodology is rather novel and important. The demonstration of an allosteric regulation mechanisms in the histone demethylase family (similar mechanism was previously reported for a DNA methyltransferase) is also interesting. Therefore, the paper can be accepted for publication in Nat Communications."

Reviewer #3 (Remarks to the Author):

I am satisfied with the revisions. The direct binding data and other supplementary experiments are a welcome addition.